



# Sea spray emissions from the Baltic Sea: Comparison of aerosol eddy covariance fluxes and chamber-simulated sea spray emissions

Julika Zinke[1,2], E. Douglas Nilsson[1,2], Piotr Markuszewski[1,2,3], Paul Zieger[1,2], E. Monica Mårtensson[1,4], Anna Rutgersson[4], Erik Nilsson[4], and Matthew E. Salter[1,2]

[1]Department of Environmental Sciences, Stockholm University, Stockholm, Sweden
[2]Bolin Centre for Climate Research, Stockholm University, Stockholm, Sweden
[3]Institute of Oceanology, Polish Academy of Science, Sopot, Poland
[4]Department of Earth Sciences, Uppsala University, Uppsala, Sweden

**Correspondence:** Julika Zinke (julika.zinke@aces.su.se)

**Abstract.** To bridge the gap between in situ and laboratory estimates of sea spray aerosol (SSA) production fluxes, we conducted two research campaigns in the vicinity of an eddy covariance (EC) flux tower on the island of Östergarnsholm in the Baltic Sea during May and August 2021. To accomplish this, we performed EC flux measurements simultaneously with laboratory measurements using a plunging jet sea spray simulation chamber containing local seawater sampled close to the footprint

of the flux tower. We observed a log-linear relationship between wind speed and EC-derived SSA emission fluxes, a power-law relationship between significant wave height and EC-derived SSA emission fluxes, and a linear relationship between wave Reynolds number and EC-derived SSA emission fluxes, all of which are consistent with earlier studies. Although we observed a weak negative relationship between particle production in the sea spray simulation chamber and seawater chlorophyll-$\alpha$ concentration and a weak positive relationship with the concentration of fluorescent dissolved organic matter in seawater, we did

not observe any significant impact of dissolved oxygen on particle production in the chamber.

To obtain an estimate of the size-resolved emission spectrum for particles with dry diameters between 0.015 and 10 µm, we combined the estimates of SSA particle production fluxes obtained using the EC measurements and the chamber measurements in three different ways: 1) using the traditional continuous whitecap method, 2) using air entrainment measurements, and 3) simply scaling the chamber data to the EC fluxes. In doing so, we observed that the magnitude of the EC-derived emission

fluxes compared relatively well to the magnitude of the fluxes obtained using the chamber air entrainment method, as well as the previous flux measurements of Nilsson et al. (2021) and the parameterizations of Mårtensson et al. (2003) and Salter et al. (2015). As a result of these measurements, we have derived a wind speed-dependent and wave state-dependent SSA parameterization for particles with dry diameters between 0.015 and 10 µm for low-salinity waters such as the Baltic Sea, thus providing a more accurate estimation of SSA production fluxes.

## 1 Introduction

Sea spray aerosol (SSA) is a major natural source of aerosols, produced when wave breaking entrains air into ocean surface water, which subsequently breaks up into bubbles. These bubbles rise to the surface where they burst and produce both a large





number of relatively small film drops resulting from the disintegration of the bubble film cap (for bubbles with diameters >
2 mm) and a smaller number of jet drops resulting from the collapse of the bubble cavity, which are typically larger in size
than the film drops (Woolf et al., 1987; Spiel, 1997). Along with wind speed, sea state, seawater temperature, salinity, and the
physicochemical and biological state of the ocean, have been shown to influence the production of SSA (e.g Woodcock, 1953;
Monahan et al., 1983; Bowyer et al., 1990; Nilsson et al., 2001; Mårtensson et al., 2003; Sellegri et al., 2006; Russell and
Singh, 2006; Tyree et al., 2007; Zábori et al., 2012; Modini et al., 2013; Park et al., 2014; Salter et al., 2014, 2015; May et al.,
2016; Schwier et al., 2017; Forestieri et al., 2018; Nielsen and Bilde, 2020).

SSA can have a significant impact on Earth's radiation budget by scattering incoming solar radiation directly and by acting
as cloud condensation nuclei (Schwartz, 1996; Murphy et al., 1998; Quinn et al., 1998). Although coarse-mode SSA typically
dominates mass emissions, fine-mode SSA has a more significant impact on radiative transfer because it more effectively
scatters solar radiation under clear-sky conditions (Haywood et al., 1999). In addition, submicrometer SSA plays a crucial role
in the concentration of cloud condensation nuclei (Fossum et al., 2020). Therefore, it is necessary to parameterize the entire
SSA size spectrum to obtain better estimates of climate forcing from model simulations.

Many sea spray source functions have been presented in the literature, varying more than an order of magnitude at any
given wind speed (de Leeuw et al., 2011). One reason for this may be the large number of environmental variables that impact
the SSA production process. For instance, while SSA production has traditionally been parameterized as a function of wind
speed, recent studies have attempted to include the impact of seawater temperature (e.g. Monahan et al., 1986; Gong, 2003;
Mårtensson et al., 2003; Clarke et al., 2006; Long et al., 2011; Kirkevåg et al., 2013; Ceburnis et al., 2014; Salter et al., 2015).
This is because wind-driven wave breaking alone is insufficient to explain the variability of SSA production estimates. In fact,
Liu et al. (2021) have demonstrated that accounting for seawater temperature enhances the predictability of observed SSA
production compared to using wind speed alone. The sea state is another environmental variable that likely influences SSA
emissions. Recent work suggests that the sea state, i.e. significant wave height or wave Reynolds number, is likely a better
predictor of SSA emissions than wind speed alone (Norris et al., 2013; Ovadnevaite et al., 2014; Yang et al., 2019).

The large variability in sea spray source functions found in the literature may also be due to the different approaches used
to derive them. Three types of approaches have been used to estimate SSA emissions. The first approach uses laboratory
experiments to mimic the wave-breaking process (e.g. Monahan et al., 1982, 1994; Mårtensson et al., 2003; Keene et al., 2007;
Tyree et al., 2007; Long et al., 2011; Salter et al., 2015). The second approach involves direct measurements of the ambient
marine atmosphere using micrometeorological techniques such as eddy covariance (e.g. Nilsson et al., 2001; Geever et al.,
2005; Norris et al., 2008, 2012; Yang et al., 2019; Nilsson et al., 2021) or the gradient method (e.g. Markuszewski et al., 2020).
The third approach is the combination of ambient aerosol concentration measurements and source-receptor modelling (e.g.
Ovadnevaite et al., 2014; Grythe et al., 2014).

Many studies estimate SSA emission fluxes indirectly using laboratory experiments. In these experiments, SSA is generated
under controlled conditions using wave chambers (e.g. Monahan et al., 1982), plunging jets of water (e.g. Salter et al., 2015),
or by forcing air through diffusers or sintered glass filters below the water surface (e.g. Mårtensson et al., 2003; Keene et al.,
2007; Tyree et al., 2007). In most studies that have attempted to derive a source function from laboratory measurements, SSA





number concentrations are converted to the size-dependent SSA production flux per whitecap area, which is then multiplied by the whitecap fraction that depends on the wind speed (Monahan and O'Muircheartaigh, 1980). However, it remains unclear

how well laboratory experiments represent the wave-breaking process, and the limited scale of the systems may introduce artifacts such as wall effects. Furthermore, accurately determining the whitecap fraction in laboratory SSA simulation chambers is challenging, hindering up-scaling the production fluxes obtained in laboratory experiments to real-world conditions. To overcome this challenge, several studies have attempted to use the volume of air entrained to scale SSA particle production fluxes obtained in laboratory systems (e.g. Long et al., 2011; Salter et al., 2015).

In contrast to indirect laboratory approaches, the eddy covariance (EC) method provides direct estimates of vertical turbulent aerosol fluxes (Buzorius et al., 1998). However, relatively few studies have used the EC method to estimate SSA emission fluxes over the open sea (Nilsson et al., 2001; Geever et al., 2005; Norris et al., 2008, 2012; Yang et al., 2019). Although this approach has the advantage of directly quantifying SSA emission fluxes, a major drawback is the requirement for aerosol instrumentation capable of fast response and sampling rates. Since optical particle counters (OPCs) are currently the only

fast-response aerosol instruments that provide size-resolved measurements, the EC method is limited to the particle size range covered by these instruments, typically $D_p > 0.1$ μm (where $D_p$ denotes the dry particle diameter). Thus, obtaining size-resolved SSA fluxes across the full-size spectrum relevant to SSA emissions using the EC method remains challenging. In addition, another drawback of the EC method is that it cannot provide information on the chemical and microbial properties of the aerosols and therefore cannot quantify the emission flux of bacteria associated with SSA, for example.

To circumvent these issues, Nilsson et al. (2021) attempted to scale laboratory-derived SSA emission estimates to in situ EC SSA emissions measured at a coastal sampling site in the Baltic Sea. They obtained a wind speed-dependent SSA emission flux over the particle size range of $0.01 < D_p < 2$ μm. However, as their dataset had only a limited number of data points from an open-sea sector, fluxes from sectors with short fetch and shallow waters had to be included. As these sectors were likely affected by coastal wave breaking, the usefulness of their measurements in understanding open-sea SSA emissions is likely

limited. To address this issue, we conducted two field campaigns in the Baltic Sea. During these campaigns, we conducted EC flux measurements on the island of Östergarnsholm and simultaneously performed measurements using a laboratory sea spray simulation chamber filled with fresh seawater collected within the flux footprint area. Our EC analysis focused on sectors representing open-sea conditions. Combining these two approaches allowed us to directly quantify the magnitude of the SSA flux and extend our emission estimates below the lower particle size limit of the OPC, obtaining wind speed- and wave-state-

dependent SSA emission fluxes over the particle size range of $0.015 < D_p < 10$ μm.

## 2   Measurement site and methods

To estimate SSA production fluxes using both in situ EC measurements and a laboratory sea spray simulation chamber, we conducted co-located ship-based experiments near the Integrated Carbon Observation System (ICOS) station on the island of Östergarnsholm in the Baltic Sea (57°25'48.4" N, 18°59'02.9" E). We carried out two campaigns: The first using the Polish

research vessel *Oceania* in May 2021, and the second using the Swedish research vessel *Electra* in August 2021. During both





campaigns, the ships were stationary in the proximity of the EC flux tower's flux footprint area. To avoid disturbing the EC flux measurements during the first campaign, R/V *Oceania* was anchored approximately 5 km away from the station. This distance kept the ship outside the flux footprint. During the second campaign, R/V *Electra* was anchored near the island, in a wind sector influenced by the presence of Gotland. We excluded data from this sector since it could have affected our measurements.

Extensive studies have been conducted on the footprint of the EC flux tower, and Rutgersson et al. (2020) identified an open-sea sector from $80°$ to $220°$ south of the station that has an undisturbed wave field without bottom topography or coastal features. In this sector, the ocean depth rapidly increases to deeper than 20 m. However, in the sector north of $80°$, the water is shallower, and the bottom topography is likely to influence SSA emissions. Similarly, SSA emissions in the sector west and north of $220°$ are likely to be influenced by the presence of Gotland and the island of Östergarnsholm, and the bottom

topography is likely to affect the wave field properties when the wave period is high. Therefore, in our data analysis, we only used aerosol EC fluxes obtained within the 80-220° sector. Figure 1 depicts the location of the flux station and the ships, as well as the wind sectors identified by Rutgersson et al. (2020).

## 2.1    EC aerosol flux tower

Stockholm University has installed a 10 m  tower for measuring aerosol EC fluxes adjacent to the 30 m ICOS mast, as shown in

Fig. 1, which is used to measure EC greenhouse gas fluxes. A horizontal head ultrasonic anemometer (Gill HS, Gill Instruments Ltd, UK) was placed on a platform at the top of the aerosol flux tower, with the 3-axis sonic head 12 m above the average sea surface level. Wind speed in three dimensions ($u$,$v$,$w$) and the atmospheric temperature derived from the speed of sound ($T_{\text{air}}$) were recorded at 20 Hz. The open side of the horizontal head faced south to maximize the quality of measurements in the open-sea sector 80–220°.

Approximately 20 cm behind the ultrasonic head, a high-speed open path $H_2O$ and $CO_2$ sensor was mounted, recording at 20 Hz (Licor 7500, Li-Cor Environmental Ltd, UK). It was placed at a slight angle to facilitate precipitation drainage from the optical surface. The Licor 7500 was zeroed using nitrogen gas, and $CO_2$ and $H_2O$ spans were calibrated using an AGA $CO_2$ gas bottle and a dew point generator (LI-610, Li-Cor Environmental Ltd, UK), respectively.

Ambient air was sampled vertically downward through a 5 m long stainless steel 1/4" sampling line, which was mounted on

a second platform at 7 m height. To prevent precipitation from entering the sampling line, a $180°$ bend was installed at the top of the sample line. The inlet was located approximatly 20 cm behind the anemometer head, to the left of the Licor 7500. The sampling line led to an OPC (Model 1.109, Grimm Aerosol Technik GmbH, Germany) that sampled at $1.2\,\text{L}\,\text{min}^{-1}$. The OPC was calibrated by Grimm Aerosol Technik GmbH, and its first-order response time (see Section A1.2) was measured at the Department of Environmental Science, Stockholm University. The OPC was set to count the aerosol number concentration $N_i$

in 15 size classes $i$ with diameters $0.25 < D_p < 2.5\,\mu\text{m}$, with a time resolution of 1 s. Given the flow through the sample line and the dimensions of the tube, the flow in the sampling line should have been laminar ($Re = \frac{4Q}{\pi D_{\text{tube}}\nu} \approx 284$, where $Q$ is the sampling flow of the OPC, $D_{\text{tube}}$ is the diameter of the sampling line, and $\nu$ is the kinematic viscosity).

Because access to the sampling site is limited and the amount of electrical power at the site is restricted, it was not possible to dry the aerosol sample. Therefore, the OPC conducted all measurements at ambient temperature and humidity. All instruments



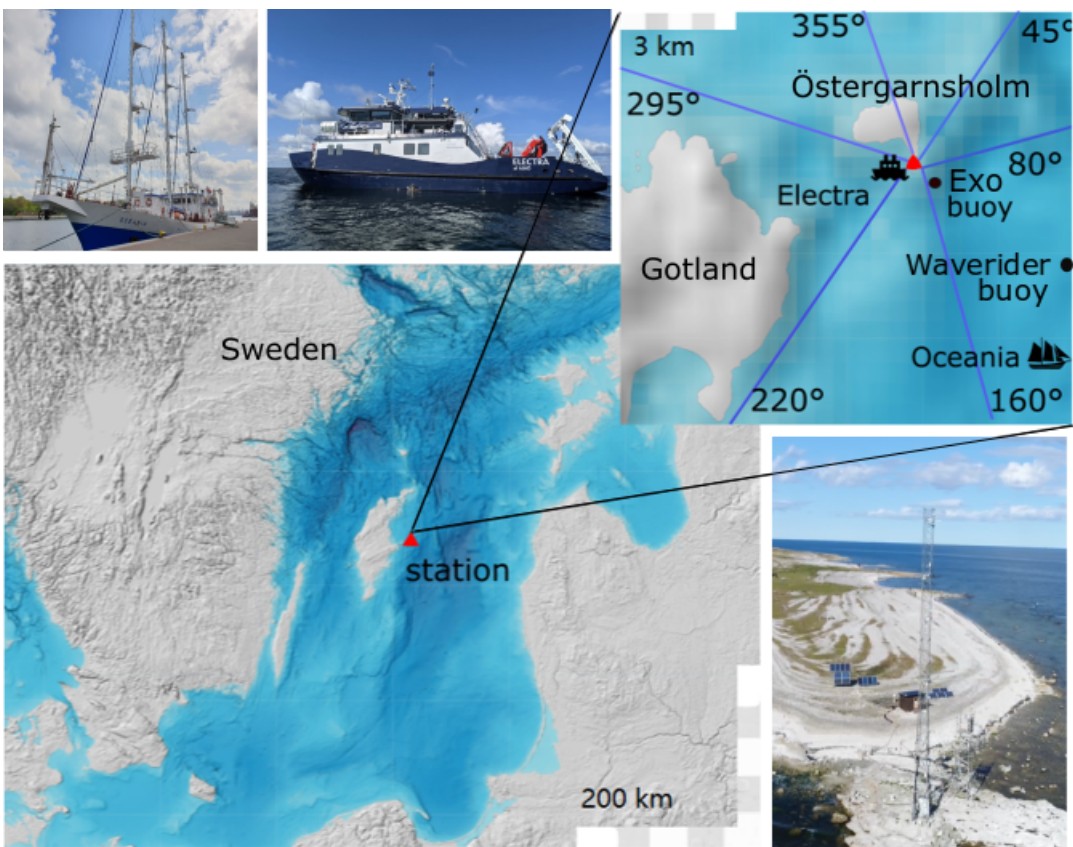

**Figure 1.** This map shows the location of the EC flux tower on Östergarnsholm island (red triangle) and the research vessels, along with the positions of the EXO2 multiparameter sensor and waverider buoy. Reference pictures of the flux tower are also included. Wind sectors are identified based on the classification by Rutgersson et al. (2020). Map © BSHC.

were recorded and monitored using a gateway and PC running the LabVIEW software SCOL-EC, developed by Stockholm University (Nilsson et al., 2021).

## 2.2 EC method and calculations

To estimate aerosol fluxes using the EC method, high-frequency measurements of aerosol number concentration are correlated with the vertical wind speed $w$. These measurements are averaged over time, typically 30 min intervals, to obtain the total and size-resolved net aerosol flux ($\overline{N'_{\text{total}}w'}$ and $\overline{N'_i w'}$), represented here using overlines and primes (') to denote the 30-minute means and turbulent fluctuations, respectively. The net aerosol fluxes are a result of transport caused by both upward motions (emission fluxes) and downward eddy motions (deposition fluxes). However, only emissions from sources within the flux footprint will contribute to upward fluxes. Aerosol particles that originate outside the flux footprint will not have a positive correlation with the vertical wind component $w$, and thus will not contribute to upward fluxes. Instead, they will contribute



to downward fluxes through dry deposition. Therefore, by estimating the dry deposition flux and subtracting it from the net aerosol flux, it is possible to derive the SSA emission flux.

The CALCEDDY LabVIEW program, which was developed at Stockholm University (Nilsson et al., 2021), was used for eliminating spikes, rotating coordinates, detrending data, correcting for lags and calculating covariances, averages and standard deviations.

## 2.3   EC footprint

In the simplest terms, the flux footprint refers to the area that the instruments on the tower "see". It represents the area upwind of the tower within which aerosol fluxes are detected by these instruments. Under stationary conditions, the footprint represents the area from where the measured fluxes originate, whether they are fluxes of momentum, heat, gases, or aerosols. The size of the footprint depends on various factors, such as the measurement height $z_m$, atmospheric stability $\frac{z_m}{L}$, the friction velocity

$u^*$, and the wind direction. Several methods can be used to determine the footprint. In the case of Östergarnsholm, it has been thoroughly studied using backward dispersion modelling (Smedman et al., 1999) and flux footprint modelling (Gutiérrez-Loza et al., 2022). According to Högström et al. (2008), for measurements taken at a height of 10 m above the surface, 80 % of the fluxes originated 800 m upwind of the tower for unstable atmospheric stability conditions, 1500 m upwind of the tower under neutral atmospheric stability conditions, and 6500 m upwind of the tower for stable atmospheric conditions.

## 2.4   EC aerosol flux errors

In order to quantify aerosol fluxes using the EC approach, we need to consider potential measurement errors resulting from physical phenomena, instrument problems, and the specifics of our particular setup. Although there are a number of potential flux errors, many can be prevented, minimized, or corrected. In this section, we introduce the different corrections we have applied to process our data. Further detail can be found in Section A of the appendix.

We distinguish between two types of errors: random stochastic errors ($\epsilon$) and systematic errors ($\delta$). For most systematic errors, there are established methods to estimate the error, which allows us to correct the measurements. However, for random errors, we can only estimate the magnitude ($\epsilon$) using statistical relationships. In the following, we provide a description of the errors that we have quantified, with an emphasis on aerosol flux errors.

### 2.4.1   Systematic EC aerosol flux errors

Systematic errors can result in a fixed bias, a relative bias that scales with the magnitude of what is being measured, or a bias that varies over time. In the EC flux system used in this study, the lateral separation between the sonic anemometer and the OPC results in a negligible error ($\delta_{ls}$), especially considering that the OPC data were only recorded at 1 Hz. We calculated aerosol EC fluxes for 30 min periods, which is a standard approach in many EC studies (e.g. Nilsson et al., 2001, 2021; Geever et al., 2005; Mårtensson et al., 2006; Ahlm et al., 2010). Although a low-cut frequency correction can be applied to account for

very large eddies that are not completely sampled during 30 min periods, this issue is more likely to occur over continental sites



during very unstable conditions. Since our data set was obtained in the marine boundary layer under close to neutral conditions, this is unlikely to be a problem, and we have not applied this correction.

Differences in the properties of the footprint in the sectors surrounding the mast can also cause errors when the instantaneous wind direction changes during the 30 min flux periods. However, we will only consider data from the open-sea sector in our

analysis, assuming that the surface properties of the footprint in this sector are fairly consistent.

Other systematic errors are large enough that we need to try to quantify them and correct the observed flux for these errors. These include the error introduced by flux losses at high frequency in closed-path systems, which is often referred to as low-pass filtering, i.e. signal damping in the sampling line to the OPC and the limited response time of the OPC. We also need to consider the effect of density fluctuations (Webb correction) and aerosol deliquescence (Kowalski correction).

A short summary of systematic errors, along with their estimated magnitudes, is provided below in the order in which they were estimated and corrected. Table 3 in Section 3.2 (see also Fig. S1) summarizes the results. For a more detailed description of the error estimation and corrections, refer to Section A in the Appendix.

The largest source of error results from the attenuation of turbulent fluctuations in the sampling line and the OPC, which leads to an underestimation of the EC flux that is constant across all particle sizes. This error is significant, corresponding to

20.7 % of the observed EC flux for the entire OPC size range (see Table 3 and Fig. S1).

The impact of the limited response time of the OPC was estimated on the basis of Horst (1997) and is constant across all particle sizes. To include the smallest eddies in EC fluxes, instruments capable of high-frequency measurements (10–20 Hz) are required. However, since the OPC used in this study was only capable of making measurements at 1 Hz, there is a substantial attenuation of the flux in our measurements that is constant with size. When normalized to the average total number flux, this

error corresponds to 13.5 % of the total aerosol number flux (see also Fig. S1 and Table 3).

Losses of aerosols due to particle diffusion, impaction, and sedimentation within the sampling line can lead to an under-estimation of the measured aerosol number concentration that varies with particle size. To estimate these losses, we used the Particle Loss Calculator, PLC, developed by von der Weiden et al. (2009). Since we minimised the bends and the length of the sampling line to the OPC and since most of the particles measured by the OPC fall within the accumulation mode, the losses

in the sampling line were relatively small. When normalized to the average total number flux, the errors due to losses in the sampling range from 0.8 % for the smallest OPC size bin centered at $D_p = 0.265$ μm to 1.5 % for the largest OPC size bin centered at $D_p = 2.24$ μm (see Fig. S1 and Table 3).

To account for the influence of water vapor fluxes on scalar concentrations of interest relative to total moist air, a correction known as the Webb correction is required (Webb et al., 1980). However, this correction is negligible in the case of the EC

aerosol fluxes measured in this study, with values ranging from 0.005 % for the smallest OPC size bin to 0 % for the largest size bin (see Table 3 and Fig. S1). One possible explanation for the small magnitude of this correction is the damping of density fluctuations in the sampling line.

To account for fluctuations in the ambient water vapor saturation ratio that can grow particles smaller than the detectable size range of the OPC into larger particles that the instrument can detect, a size-dependent correction must be applied. We used



the approach of Kowalski (2001) to estimate a relative error of $0.28\,\%$ in the smallest size bin and $0\,\%$ in the largest size bin compared to the uncorrected EC flux (see Table 3 and Fig. S1).

### 2.4.2  Random EC aerosol flux errors

Random errors are dependent on the sample size, and as such, a higher number of data points result in smaller random errors because they average out. When calculating EC fluxes, it is essential to consider a number of random errors, including variations

in the prevailing wind direction and resulting differences in footprint properties during the 30 min averaging periods. However, as mentioned previously, we only considered fluxes obtained during periods when the wind blew from the open-sea sector, assuming that the footprint surface properties were similar across this sector.

For particle-counting instruments such as OPCs and CPCs, a fraction of the random error is related to the discrete counting of the particles. The discrete counting error increases with increasing particle size and decreasing particle concentration. In the

case of this data set, the discrete counting error accounted for a relative flux error of 43-48 % (maximum $\sim 75\,\%$ at $D_p = 1.5$ µm). Unlike systematic errors, random errors cannot be corrected, and instead, we will indicate them as error bars in the following data analysis. Generally, random errors were limited to less than $10 - 20\,\%$ of the EC fluxes.

### 2.5  Estimation of sea spray aerosol emission fluxes using an aerosol dry deposition model

To estimate the actual SSA emissions, we need to model the dry deposition fluxes and subtract them from the net aerosol fluxes.

To do so, we use:

$$EF_i = \overline{N_i'w'} - \overline{N_i} \times v_d(D_i) \tag{5}$$

Here $v_d$ is the size-dependent aerosol dry deposition velocity for each size bin diameter $D_i$, following the approach of Nilsson et al. (2001) and Nilsson et al. (2021). We use the parameterization of dry aerosol deposition by Schack Jr et al. (1985) for $v_d$, set for the wind tunnel parameters corresponding to water surfaces at $u* = 0.44\,\mathrm{ms}^{-1}$. Therefore, the emission flux for

the entire OPC size range is:

$$EF_{\text{total}} = \sum_{(i=1)}^{15} \left( \overline{N_i'w'} - \overline{N_i} \times v_d(D_i) \right) \tag{6}$$

### 2.6  Spectral analysis

To identify EC data points that should be excluded from the analysis, we calculated the turbulence power- and co-spectra using a fast Fourier transform (FFT) for each 30 min time period. The power- and co-spectra were frequency-weighted and

normalized by the variance or the covariance, respectively. We excluded a 30 min period if the slope of the power spectrum deviated notably from -2/3 on the normalized scale in the inertial sub-range or if the slope approached +1 (white noise) at a frequency lower than the expected response time of the instrument. Similarly, we excluded a 30 min period if the slope of



the co-spectra deviated notably from the -4/3 slope. We divided the 30 min time periods into three categories: A) good data,
B) non-ideal data, and C) poor data. An example of the power and co-spectra for aerosol, temperature, horizontal wind speed

and water vapor fluxes for "good data" is presented in Fig. S2 in the supplement. As can be seen in Fig. S3, the impact of the
spectral analysis on the size-resolved fluxes was small. In the following analysis, we used only data from 30 min periods that
were classified as "good".

## 2.7   Production of nascent SSA using a laboratory sea spray simulation chamber

A laboratory sea spray simulation chamber was used to generate nascent SSA during two research cruises in the vicinity of

Östergarnsholm. R/V *Oceania* was stationed there from 19 May 2021 at 16:00 to 22 May 2021 at 00:00 (local time, LT), and
again from 23 May 2021 at 00:00 to 24 May 2021 at 04:00 LT. R/V *Electra* was was also in the area from 10 August 2021 at
09:30 until 22 August 2021 at 08:00 LT, but had to leave its anchored position on 16 August at 08:00 LT to return to the harbor
in Fårösund for refueling due to poor weather conditions. The ship returned to Östergarnsholm on 18 August at 08:00 LT but
it was not possible to anchor in the same position, and the ship had to return to the nearby harbour each evening (17:00-08:00

LT) until the end of the campaign. Therefore, in the following sections, we only include chamber measurements obtained when
the ship was located close to the station on Östergarnsholm.

The sea spray simulation chamber used for the experiments is described in detail in Salter et al. (2014). In summary, SSA
particles were generated by a plunging jet that hits the water surface from a height of 40 cm, entraining air into the water. The
entrained air rises in the form of bubbles that burst and expel droplets, which are eventually dried and sampled into aerosol

instrumentation. The sea spray simulation chamber operates under a slight positive pressure by introducing particle-free sweep
air to exclude the possibility of outside air contamination and to ensure that the headspace of the chamber is well mixed.
Although the chamber can be temperature controlled, it was operated without temperature control in this study because the
seawater in the chamber was constantly being replaced and thus was at ambient temperature. This makes our experiments
comparable to previous chamber experiments that used a plunging jet and fresh seawater (e.g. Facchini et al., 2008; Hultin

et al., 2010, 2011; Zábori et al., 2012, 2013). The chamber was continuously filled with local surface seawater sampled using
the seawater inlets of the ships. During the R/V *Oceania* campaign, inline measurements of seawater temperature, $T_{\mathrm{seawater}}$,
and salinity, $S$, were made using a seabird CTD probe (SBE 21 SeaCAT Thermosalinograph, Sea-Bird Scientific, USA) and
oxygen saturation was measured with an oxygen meter (Fibox 4 trace, PreSens Precision Sensing GmbH, Germany). During
the R/V *Electra* campaign, the seawater temperature in the chamber was continuously measured using a conductivity sensor

(model number 4120, Aanderaa, Norway) and the dissolved oxygen (DO) concentrations in the chamber were measured with
an oxygen optode (model number 4175, Aanderaa, Norway). The concentrations of chlorophyll-$\alpha$ and fluorescent dissolved
organic matter (FDOM) were measured inline with two fluorometers (Cyclops-7F, Turner Designs, USA). Additionally, we
utilised salinity data measured by an EXO2 multiparameter sensor (YSI Inc., Yellow Springs, OH, USA) installed on a mooring
1 km south east of the station by Uppsala University. Measurements of wave properties were made with a Directional Waverider

260  moored at a depth of 39 m, 4 km southeast of the tower. For more details on the wave measurements, we refer the reader to
Rutgersson et al. (2020) and Hallgren et al. (2022).



### 2.7.1 Measurements of the aerosol size distribution

The size distribution of the aerosols produced in the chamber was measured using a custom-built differential mobility particle sizer (DMPS), which consisted of a Vienna-type differential mobility analyzer (DMA) and a condensation particle counter (CPC, model 3772, TSI, USA) with a flow rate of $1\,\mathrm{L\,min^{-1}}$, that measured particles with electrical mobility diameters between 0.015 and 0.906 µm distributed over 37 size bins. We also used a white-light optical particle size spectrometer with a flow rate of $5\,\mathrm{L\,min^{-1}}$ (WELAS 2300 HP sensor and Promo 2000 H, Palas GmbH, Germany, hereafter called WELAS), which measured particles with optical diameters between 0.150 and 10 µm distributed over 59 bins. To combine the size distributions measured by the DMPS and WELAS, we have converted the optical diameters measured by WELAS to volume equivalent diameters assuming a refractive index of $m = 1.54 - 0i$ for sea salt particles, which corresponds to the value of NaCl (Abo Riziq et al., 2007). We carried out the conversion using the software provided by the manufacturer (PDAnalyze Version No. 2.024, Palas GmbH, Karlsruhe, Germany), which was based on instrument-specific Mie calculations. The diameters of the aerosol particles were also shape corrected according to Zieger et al. (2017). Before sampling, we dried the particle-laden air in two Nafion dryers (model MD-700-36F/48F, Perma Pure, USA) that were horizontally mounted in front of the DMPS and WELAS. We monitored the temperature and relative humidity (RH) of the sample with two sensors (HYTELOG-USB, B+B Thermo-Technik GmbH) mounted in front of the sampling inlets of the WELAS and DMPS system to ensure that the measured particle diameters could be considered dry diameters. The average RH (measured behind WELAS) was $31.7 \pm 2\%$ for the Oceania campaign and $18.9 \pm 1.6\%$ during the Electra campaign (mean $\pm$ standard deviation).

To estimate losses in the sampling lines we used the Particle Loss Calculator Software (von der Weiden et al., 2009). After correcting for all factors, we combined the DMPS and WELAS data at measured particle sizes of 0.35 µm. All sizing instruments were calibrated with polystyrene latex spheres.

### 2.7.2 Derivation of SSA production fluxes from the chamber measurements using the continuous whitecap method

To estimate the production flux of SSA particles using chamber measurements, we employed the continuous whitecap method (CWM, e.g. Cipriano and Blanchard, 1981; Mårtensson et al., 2003). The CWM combines an estimate of the size-resolved number of SSA particles produced per unit of whitecap area per second in the chamber with an estimate of whitecap coverage to predict the size-resolved interfacial number of SSA particles per unit of ocean surface area per unit of time (Lewis and Schwartz, 2004).

One of the most widely used sea spray source functions is based on the discrete whitecap method (DWM, Monahan and O'Muircheartaigh, 1980). This source function combines laboratory experiments that measured the size-resolved number of SSA particles produced by a simulated breaking wave and the oceanic whitecap coverage ($W$), which is often parameterized in terms of the wind speed at 10 m above the sea surface ($U_{10\,\mathrm{m}}$). For example, Monahan and O'Muircheartaigh (1980) used the following empirical relationship:

$$W = 3.84 \times 10^{-4} \cdot U_{10\,\mathrm{m}}^{3.41}. \tag{7}$$





It is important to note a key difference between the CWM we used and the DWM developed by Monahan and colleagues
(e.g. Monahan and O'Muircheartaigh, 1980). The goal of the DWM, as originally formulated, was not to determine the number
of SSA particles produced per unit of whitecap area per second. Instead, this approach aimed to determine the number of SSA
particles produced per unit whitecap area from a laboratory-simulated breaking wave over the entire lifetime of the resulting
whitecap and associated degassing bubble plume. See Callaghan (2013) for a detailed discussion of this.

To calculate the SSA production flux $\frac{\mathrm{d}F}{\mathrm{dlog}D_\mathrm{p}}$, we multiply the flux per whitecap area by the whitecap coverage using the
following equation:

$$\frac{\mathrm{d}F}{\mathrm{dlog}D_\mathrm{p}} = \frac{\frac{\mathrm{d}N}{\mathrm{dlog}D_\mathrm{p}} \cdot Q_\mathrm{sweep}}{A_\mathrm{surface}} \cdot W. \tag{8}$$

where $\frac{\mathrm{d}N}{\mathrm{dlog}D}$ is the measured size distribution, $Q_\mathrm{sweep}$ is the sweep flow, and $A_\mathrm{surface}$ is the seawater surface area inside the
chamber covered by bubbles. However, our experimental setup has a limitation: we did not determine the exact surface area of
seawater covered in bubbles. Instead, we assumed that the entire surface of the seawater was covered in bubbles, and used the
total surface area of the seawater.

### 2.7.3 Derivation of SSA production fluxes from the chamber measurements using air entrainment

Another method for obtaining estimates of the production flux of SSA particles from breaking waves and whitecaps using sea
spray simulation chambers has been developed by Long et al. (2011) and Salter et al. (2015). These authors combined the
number of particles produced per unit time in a logarithmic interval of $D_p$ with measurements of air entrainment/detrainment.
This approach assumes that all air entrained into the water column detrains as bubbles that produce particles, and does not
consider other factors that may affect the air entrainment flux, such as breaking wave strength or sea state.

To apply this approach, we measured the volume of air entrained in a manner similar to Salter et al. (2014) under conditions
relevant to our field measurements, using seawater from the footprint area ($S = 6\,\mathrm{g\,kg^{-1}}$ and $T_\mathrm{seawater} = 10°\mathrm{C}$ and $T_\mathrm{seawater} = 20°\mathrm{C}$, respectively, for the May and August campaigns). To measure the volume of air entrained by the plunging jet, we
enclosed the jet in a stainless steel tube, with the base of the tube submerged $10\,\mathrm{mm}$ below the seawater surface, and recorded
the volumetric air flow 30 times using a flow meter (Gillibrator 2, Sensidyne, USA). Using these estimates of air entrainment,
$Q_\mathrm{air}$, we can estimate the particle production rate $f$ (per $\mathrm{m}^3$) as follows:

$$f = \frac{\frac{\mathrm{d}N}{\mathrm{dlog}D_\mathrm{p}} \cdot Q_\mathrm{sweep}}{Q_\mathrm{air}(T)}. \tag{9}$$

The size-resolved interfacial flux is then obtained by multiplying the particle production rate by a parameterization of the air
entrainment flux $F_\mathrm{ent} = (2 \pm 1) \times 10^{-8} \cdot U_{10\,\mathrm{m}}^{3.41}$ (Long et al., 2011):

$$\frac{\mathrm{d}F}{\mathrm{dlog}D_\mathrm{p}} = f(D_\mathrm{p}, T) \cdot F_\mathrm{ent}. \tag{10}$$



### 2.7.4 Derivation of SSA production fluxes by scaling size-resolved chamber measurements to ambient fluxes

Combining EC flux measurements with chamber measurements enables us to estimate the sea spray source across the full particle size range. To achieve this, we compared the aerosol number concentration ($N_j$) measured in the sea spray simulation chamber across WELAS size-bins ($j$) with the vertical aerosol number flux ($\overline{N_i'w'}$) from the EC flux system over size-bins ($i$) and obtained a scaling factor. Because the WELAS operating on the sea spray chamber and the Grimm OPC measuring EC fluxes on the tower are different and operate using different size-bins, we interpolated the WELAS data to the EC flux OPC data range ($0.25 < D_p < 2.5\,\mu\text{m}$) using the MATLAB spline function. This enabled us to estimate the ratio (R) of the EC flux to the concentration of particles measured in the sea spray simulation chamber (SSSC):

$$R_{\text{EF}_{\text{insitu}}:\text{SSSC}}(i) = \frac{\overline{N_i'w'}}{N_j'} \tag{11}$$

where $R_{\text{EF}_{\text{insitu}}:\text{SSSC}}$ has the unit $\text{m\,s}^{-1}$.

It is important to note that the particles produced in the chamber experiments were dried before being sized and counted by DMPS and WELAS, while the EC flux OPC measured particles at ambient $RH \approx 80\%$. For comparability, we have converted all diameters to radii at $RH = 80\%$ and referred to them as $R_{80}$, unless explicitly stated otherwise. We used only the flux measurements obtained simultaneously with the chamber experiments to scale the chamber data.

## 3 Results and Discussion

### 3.1 Synoptic scale and micrometeorological overview

Figure 2 displays a time series of micrometeorological and synoptic parameters including wind speed, direction, friction velocity, air temperature, and RH, as well as ambient aerosol concentrations and net fluxes from all sectors. The fluxes from the coastal-influenced sector, which were excluded from the analysis, are indicated in gray. Upward aerosol fluxes dominated during both campaigns, with 475 half-hour periods being dominated by upward aerosol fluxes and 150 half-hour periods being dominated by downward aerosol fluxes across both campaigns.

Table 1 provides an overview of synoptic scale atmospheric and seawater properties including wind speed and direction, fetch, atmospheric pressure, air and seawater temperature, salinity, concentration of dissolved oxygen and chlorophyll-$\alpha$ in seawater, during the two campaigns. Frequency histograms of these parameters for both campaigns are also presented in Figs. S4 and S5. Additionally, wind roses of the prevailing wind directions and wind speeds during both campaigns are shown in Fig. S6. Measurements at the site indicated that during both campaigns, the air mainly came from the west to south-west, accounting for 85 % of the measurement time. Back-trajectories for both campaigns were calculated with the HYSPLIT model (Stein et al., 2015; Rolph et al., 2017) and are presented in Fig. S7 and S8 in the supplementary material.

During periods with southerly winds, the distance the sampled airmass spent above open water ranged from 500 to 1200 km, with the highest values observed during the Electra campaign. The local wind speed $\overline{U}$ (averaged over 30 min intervals) ranged



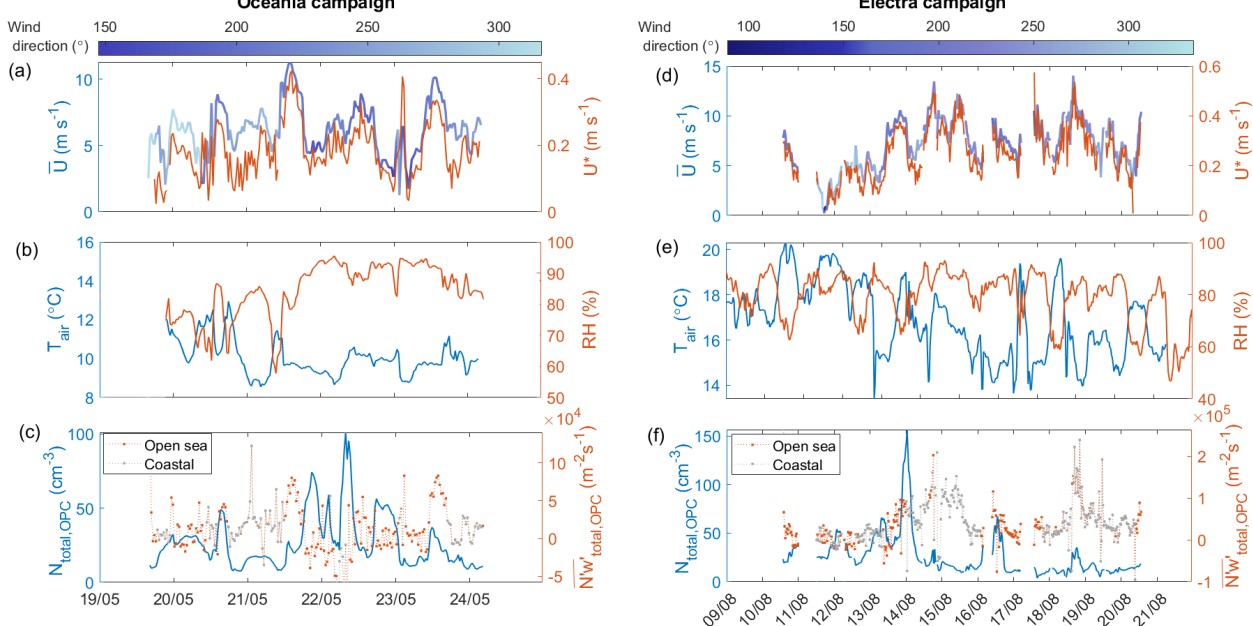

**Figure 2.** This figure shows time series of relevant micrometeorological and aerosol parameters for the Oceania and Electra campaigns. Panel (a) displays wind speed and direction, as well as friction velocity for the Oceania campaign. Panel (b) displays air temperature and relative humidity for both campaigns. Panel (c) shows the ambient particle concentration ($N_{\text{OPC}}$) and net fluxes measured on Östergarnsholm during the Oceania campaign. Panels (d) to (f) show the same parameters as panels (a) to (c) but for the Electra campaign.

from 0 to $14\,\text{m s}^{-1}$ but was mostly between 4 and $10\,\text{m s}^{-1}$ (see Fig. 2 and Table 1). A clear seasonal increase in seawater and air temperature was observed between May and August. As a proxy for phytoplankton biomass, chlorophyll-$\alpha$ (CHL) was used. The growth of plankton biomass in the Baltic Sea is characterized by three distinct peaks in the annual cycle (Wasmund

et al., 1996; Stoń-Egiert and Ostrowska, 2022): the first in spring (March/April), when diatoms and then dinoflagellates bloom; the second in summer (July/August) when cyanobacteria bloom; and the third in autumn (September/October) when diatoms bloom again. The Oceania campaign, which took place in May, occurred between the spring and summer blooms, while the Electra campaign, which took place in August, occurred towards the end of the summer bloom. Therefore, it is not surprising that higher levels of chlorophyll-$\alpha$ were measured during the Electra campaign (see Table 1).



**Table 1.** Overview of synoptic-scale atmospheric and seawater properties during the two campaigns, presented as mean and standard deviation of values.

| | Atmospheric parameters | | | | |
|---|---|---|---|---|---|
| | $T_{\mathrm{air}}$ (°C) | RH (%) | $\overline{U}$ (m s$^{-1}$) | WD (°) | $p$ (hPa) |
| Oceania campaign, May | 10.4±1.4 | 80.6±9.9 | 6.4±1.9 | 229±53.8 | 1002±3.3 |
| Electra campaign, August | 17.4±1.6 | 80.6±8.0 | 6.6±2.7 | 250.7±44.4 | 1003±7.4 |
| | Seawater properties | | | | |
| | $T_{\mathrm{seawater}}$ (°C) | $S$ (g kg$^{-1}$) | DO (μM L$^{-1}$) | CHL (mg m$^{-3}$) | Fetch (km) |
| Oceania campaign, May | 9.8±0.4 | 7.0±0.01 | 379±15.8 | 3.9±0.3 | 264±264 |
| Electra campaign, August | 18±1.1 | 6.7±0.1 | 282±3.6 | 5.2±1.1 | 188±298 |

Table 2 and the histograms in Fig. S9 provide an overview of the micrometeorological conditions encountered during both campaigns in the open sea sector. The mean stability was close to neutral during both campaigns with values of -0.02 ± 0.18 during the May campaign and -0.06 ± 0.12 during the August campaign. Stability affects the turbulent exchange of heat and water vapor, where unstable conditions lead to enhanced turbulence and stable conditions suppress turbulent exchange (see also Svensson et al. (2016) for stratification characteristics). This is also reflected in the latent and sensible heat fluxes. Sensible

heat fluxes in the open sea sector were close to zero during the May campaign (0.22 ± 10.1 W m$^{-2}$) and upward during the August campaign (7.02 ± 14.14 W m$^{-2}$). The latent heat fluxes were higher in August than in May (11.9 ± 13.9 W m$^{-2}$ in the open sea sector in May compared to 41.1 ± 21.6 W m$^{-2}$ in August), which can be explained by increased evaporation as a result of higher seawater temperatures in August. Similar patterns in stability and latent/sensible heat exchange have previously been observed at Östergarnsholm (Rutgersson et al., 2020). The mean friction velocity for the open-sea sector was 0.21 m s$^{-1}$

during the May campaign and 0.26 m s$^{-1}$ during the August campaign, which agrees well with the measurements reported in Rutgersson et al. (2020). Since variations in micrometeorological parameters in the open-sea sector were small between the two campaigns, we have combined these datasets in the analysis that follows.

**Table 2.** This table provides an overview of the micrometeorological conditions encountered during both campaigns in the open sea sector (80-220°). The values are presented as mean and standard deviation. The table includes sensible heat flux $H$, latent heat flux $\lambda E$, neutral drag coefficient ($CD_N$), wave age ($\frac{c}{U_{10\ \mathrm{m}}}$), and significant wave height ($H_s$).

| | $\frac{z}{L}$ | $H$ | $\lambda E$ | $u*$ | $z_0$ | $CD_N$ | $\frac{c}{U_{10\ \mathrm{m}}}$ | $H_s$ |
|---|---|---|---|---|---|---|---|---|
| | (-) | (W m$^{-2}$) | (W m$^{-2}$) | (m s$^{-1}$) | (m) | (-) | (-) | (m) |
| Oceania campaign | -0.02 | 0.22 | 11.88 | 0.21 | 9.85·10$^{-5}$ | 1.2·10$^{-3}$ | 1.12 | 0.59 |
| | ±0.18 | ±10.10 | ±13.94 | ± 0.09 | ±7.62·10$^{-5}$ | ±1.3·10$^{-3}$ | ±0.47 | ±0.24 |
| Electra campaign | -0.06 | 7.02 | 41.11 | 0.26 | 1.44·10$^{-4}$ | 1.2·10$^{-3}$ | 0.99 | 0.85 |
| | ±0.12 | ±14.14 | ±21.6 | ± 0.1 | ±9.09·10$^{-5}$ | ±3.73·10$^{-4}$ | ±0.32 | ±0.34 |





A description of the diurnal cycles of the ambient aerosol concentration and fluxes, as well as the micro-meteorological parameters and seawater properties mentioned above, is also provided in section B in the supplement.

### 375 3.2 Ambient aerosol concentrations and fluxes

In total, we obtained 645 half-hour estimates of the net aerosol flux by combining the data from the two campaigns, of which 232 originated from the open-sea sector. After excluding data periods characterised as non-ideal or poor, based on spectral quality control, and data points when the ships were not located close to the station, we were left with 186 30 min periods.

Figure S1b shows the size-resolved aerosol net fluxes ($\overline{N_i'w'}$) before and after applying all corrections. Additionally, it

shows the aerosol emission flux derived from the corrected net aerosol flux after subtracting the aerosol dry deposition flux.

As shown in Table 3, the median uncorrected net flux was $1.7 \times 10^4$ m$^{-2}$s$^{-1}$, which increased to $2.12 \times 10^4$ m$^{-2}$ s$^{-1}$ after applying all corrections. The estimated median total dry deposition flux was $7.86 \cdot 10^2$ m$^{-2}$s$^{-1}$, several orders of magnitude lower than the SSA emission flux ($< 4\%$), which was estimated to be $2.26 \cdot 10^4$ m$^{-2}$s$^{-1}$ (median of the integrated fluxes across all OPC size bins). Previous studies over the open sea have estimated deposition fluxes of between 14 % (Yang et al., 2019)

and 30 % of the net flux (Geever et al., 2005; Nilsson et al., 2021).

Similar to a previous study in this region (Nilsson et al., 2021), the correction for aerosol losses in the sampling line and the correction that accounts for dry deposition fluxes had only minor impacts on total and size-resolved fluxes. This is likely because the OPC mostly samples the accumulation mode, where deposition in sampling tubes or surfaces within the flux footprint is minimal.




**Table 3.** The table presents the median values of the uncorrected and fully corrected net aerosol number fluxes, along with the systematic and random aerosol errors. In addition, the table shows the modeled aerosol dry deposition flux and the estimated emission flux. The OPC bins are labeled as i=1-15.

| Aerosol EC flux | Symbol | Magnitude (median) | Relative error/correction (median) | Notes |
|---|---|---|---|---|
| Units: | | m$^{-2}$ s$^{-1}$ | % | |
| Uncorrected OPC EC flux, size resolved | $\overline{N_i'w'}$ | $1.08\times10^5$ (i=1) to $4\times10^1$ (i=15) | – | |
| total | $\overline{N_{total}'w'}$ | $1.7\times10^4$ | – | |
| Corrected OPC EC flux, size resolved | $\overline{N_i'w'}$ | $1.4\times10^5$ (i=1) to $2\times10^1$ (i=15) | - | |
| total | $(\overline{N_{total}'w'})_c$ | $2.1\times10^4$ | | |
| **Aerosol EC flux errors** | | | | |
| *Systematic flux errors (positive values correspond to underestimated fluxes = positive corrections)* | | | | |
| Fluctuation attenuation in the sampling line | $\delta_{asl}$ | $2.27\times10^4$ (i=1) to $8\times10^0$ (i=15) | 20.7 % | Size independent |
| Losses due to limited response time | $\delta_{lrt}$ | $1.6\times10^4$ (i=1) to $5\times10^0$ (i=15) | 13.5 % | Size independent |
| Particle losses in the sampling line | $\delta_{tpl}$ | $8.6\times10^2$ (i=1) to 0.6 (i=15) | 0.8% (i=1) to 1.5 % (i=15) | Size dependent, estimated with PLC* |
| Webb correction | $\delta_{Webb}$ | 8.4 (i=1) to 0.00 (i=15) | 0.005 % (i=1) to 0.00 % (i=15) | Size dependent |
| Kowalski correction | $\delta_{Kowalski}$ | $5.5\times10^2$ (i=1) to $4\times10^{-3}$ (i=15) | 0.28 % (i=1) to 0.00 % (i=15) | Size dependent |
| **Random flux errors** | | | | |
| Overall random error | $\epsilon_{or}$ | - | 10 − 20% | estimated |
| Discrete counting error | $\epsilon_{total}^{DC}$ | $9.6\times10^4$ (i=1) to $4.1\times10^2$ (i=15) | 43 % (i=1) to 48% (i=15), maximum: 75 % (i=13) | Size dependent |
| Dry deposition flux | $\overline{N_i}\times v_d(D_i)$ | $-9.4\times10^3$ (i=1) to $-4.3\times10^0$ (i=15) | 4.1 % (i=1) to 0.4 % (i=15) | Size dependent |
| Emission Estimate, size resolved | $\frac{dEF}{dlogD_p}$ | $1.6\times10^5$ (i=1)-$3\times10^1$ (i=15) | - | |
| total | $EF_{total}$ | $2.26\times10^4$ | | |

* (von der Weiden et al., 2009)



### 3.2.1 Dependence of aerosol net fluxes on the micrometeorology

The correlations between the measured net aerosol fluxes and micrometeorological parameters, such as drag coefficient, roughness length, friction velocity, stability, sensible and latent heat flux, as well as turbulent kinetic energy, are shown in Fig. S10 in the supplement. The net aerosol fluxes demonstrate positive correlations with the wind speed, roughness length, friction velocity, turbulent kinetic energy, significant wave height and wave Reynolds number, and a negative correlation with wave age. In the following sections, we will focus on the dependence of the source flux on the wind speed $U_{10\,\mathrm{m}}$, significant wave height $H_s$, and wave Reynolds number $ReH_w = \frac{u*H_s}{\nu_w}$ which was calculated based on Zhao and Toba (2001) ($\nu_w$ represents the viscosity of water).

### 3.2.2 Dependence of aerosol emission fluxes on wind speed

As shown in Fig. 3a, the SSA emission fluxes exhibit a logarithmic increase with a linear increase in wind speed:

$$\log(\mathrm{N'w'} \text{ or EF}) = a \cdot U_{10\ \mathrm{m}} + b \tag{12}$$

which is consistent with the findings of many previous studies (Nilsson et al., 2001; Geever et al., 2005; Norris et al., 2008, 2012; Yang et al., 2019; Nilsson et al., 2021).

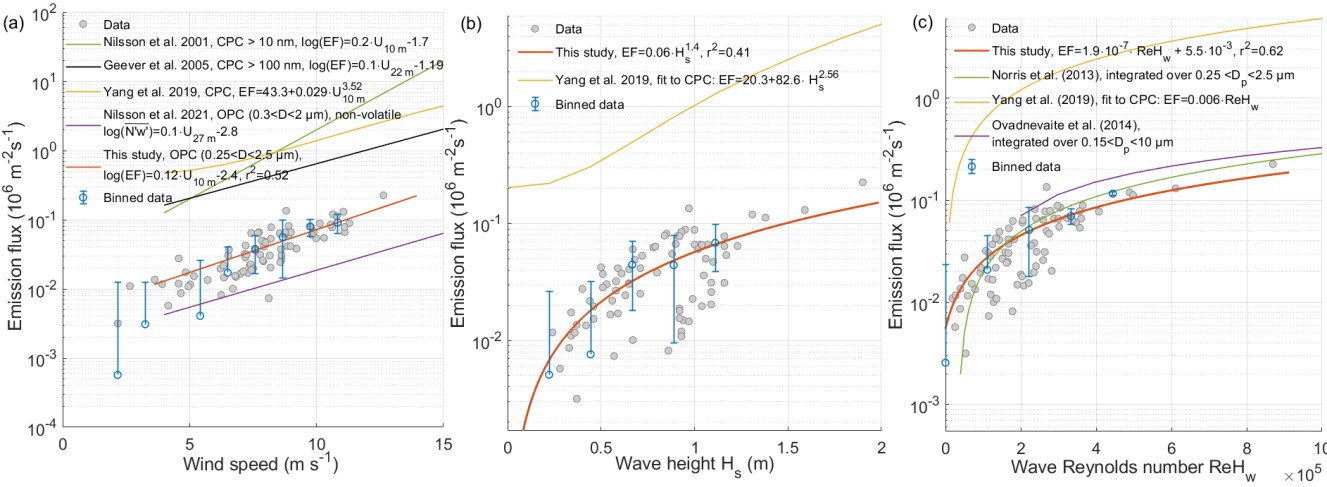

**Figure 3.** This figure presents the SSA emission flux versus (a) wind speed $\geq 4\,\mathrm{m\,s^{-1}}$, (b) significant wave height, and (c) wave Reynolds number. The grey dots represent the 30 min emission fluxes, while the blue lines represent binned data and the orange lines represent fits to the individual 30 min data periods. Additionally, we compare our results with those from previous studies.

In Fig. 3a we show the relationship between the wind speed ($U_{10\,\mathrm{m}} \geq 4\,\mathrm{m\,s^{-1}}$, the wind speed at which wave breaking starts) and the observed SSA emission fluxes in a series of studies on SSA emission fluxes over the open ocean. Our observations show a smaller magnitude of the SSA emission flux compared to most other studies by about an order of magnitude, while





the slope of the emission fluxes plotted against wind speed is reasonably comparable to that of Geever et al. (2005). There are several reasons why this may be the case. For instance, we measured SSA emission fluxes for particles with $0.25 < D_p < 2.5$ µm, while Geever et al. (2005) measured SSA emission fluxes for particles with $D_p > 100$ nm and Yang et al. (2019) and Nilsson et al. (2001) used CPCs to measure SSA emission fluxes for particles with $D_p > 1.5$ nm and $D_p > 10$ nm, respectively.

Another possible explanation is that the Baltic Sea is a semi-enclosed sea, and therefore, the conditions might not always be representative of those in open-ocean conditions. We have included the fit for the relationship between wind speed and net flux from a coastal site in the Baltic Sea reported in Nilsson et al. (2021) for comparison. It has a very similar slope of the emission fluxes plotted against wind speed as observed in our study and an even smaller magnitude, supporting our hypothesis that wind-induced fluxes are smaller in the Baltic Sea than in open-ocean conditions.

Factors such as aerosol dry deposition fluxes, boundary layer height, salinity, seawater temperature, fetch, sea ice fraction, seawater depth, wave field, and the presence of surfactants at the seawater surface can affect the slope and intercept of the fit in Fig. 3a. Furthermore, the fit parameters are dependent on particle size. Figure S11 shows that a log-linear relationship between SSA emission fluxes and wind speed can also be observed in each separate size bin of the OPC. The slopes $a$, intercepts $b$, and coefficients of determination $r^2$ for the size-resolved SSA emission fluxes are presented as a function of aerosol size in Fig.

S12. The change in slope with size provides an estimate of the number of additional particles per surface area and second that are emitted for the same change in wind speed, with the highest increase observed for particle diameters between 0.3 to 1 µm, where the correlation coefficients are highest. Since particles of this size likely originate as film drops, this indicates that film drop production is potentially more sensitive to changes in wind speed than jet drop production under the conditions in which our measurements were made. When comparing the fits of the separate size bins to the findings from Norris et al. (2008), we

note a reasonable agreement with the slopes of the fits observed in their study.

### 3.2.3 Dependence of aerosol emission fluxes on wave properties

Figures 3b and c present a comparison between the aerosol emission and two wave parameters, significant wave height $H_s$ and wave Reynolds number $ReH_w$. Binning the data into regularly distanced intervals based on the median values, reveals trends. The emission flux shows a power-law increase with increasing significant wave height, which is similar to the relationship

reported by Yang et al. (2019) (although their emission flux was more than an order of magnitude higher). Additonally, there is a linear increase in the emission flux with increasing wave Reynolds number, which agrees very well with the parameterizations by Norris et al. (2013) and Ovadnevaite et al. (2014).

In their study, Yang et al. (2019) observed that higher values of $U_{10\,m}$ and $H_s$ resulted in higher size-resolved aerosol emission fluxes across all aerosol sizes ($0.1 < R_{80} < 6\,µm$). The effect was found to be stronger for $H_s$ than for $U_{10\,m}$. In Fig.

4 we present our own findings on how size-resolved aerosol emission fluxes depend on $U_{10\,m}$, $H_s$ and $ReH_w$ and compare them with the results of Ovadnevaite et al. (2014). Note that we did not include the data from Yang et al. (2019) in Fig. 4 since their flux measurements were several orders of magnitude higher than ours, likely due to the reasons outlined earlier.

We found that our sea spray aerosol emissions, like those reported by Yang et al. (2019), are strongly influenced by the significant wave height $H_s$. Specifically, we observed a significant difference in size-resolved aerosol fluxes of between a





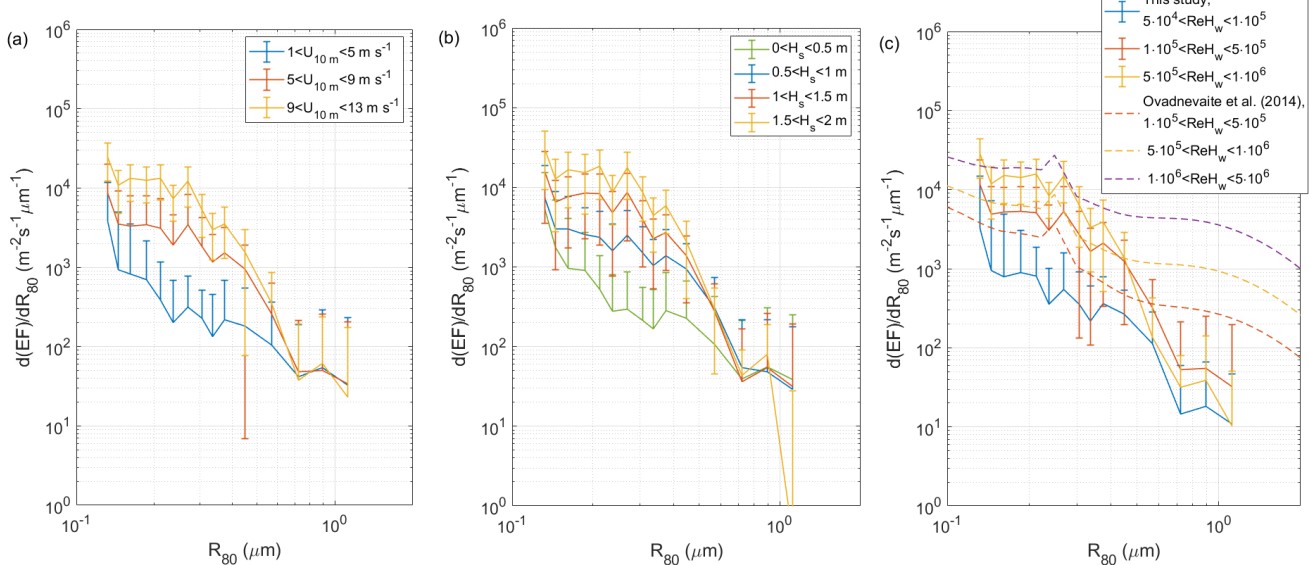

**Figure 4.** The figure shows the size-resolved emission flux dependence on (a) wind speed, (b) significant wave height and (c) wave Reynolds number, compared with the parameterization from Ovadnevaite et al. (2014).

factor of 1–4 depending on the size bin (at a probability value of $p = 0.005$ and at a significance level of 5 %) for $0.5 < H_s < 1$ m and $1 < H_s < 2$ m. Similarly, the data from Yang et al. (2019) differed by a factor of 1–5 for the same wave height ranges. Moreover, for wind speeds $U_{10\,m} < 5$ m s$^{-1}$ and $U_{10\,m} > 9$ m s$^{-1}$, we found a difference in aerosol flux of more than an order of magnitude (at a probability value of $p = 0.001$ and at a significance level of 5 %), while Yang et al. (2019) reported a much smaller difference, even over a wider range of $U_{10\,m}$.

Finally, from Fig. 4c it is apparent that the wave Reynolds number strongly affects the size-resolved aerosol emission fluxes that we observed. In this regard, our data set agrees very well with the parameterization of Ovadnevaite et al. (2014).

### 3.3 Simulated sea spray production in the chamber experiments with water from the footprint area

Figure 5 shows a time series of the particle concentration measured in the headspace of the sea spray simulation chamber, as well as flux estimates derived from the entrainment method, continuous whitecap method, and from scaling the chamber data to
in situ fluxes. The figure also includes seawater properties such as seawater temperature, salinity, chlorophyll-$\alpha$ and dissolved oxygen, which were monitored during both the Oceania and Electra campaigns. Periods when the research vessels were not anchored close to the station were excluded, while periods when the wind was blowing from outside the open-sea sector are shaded.

The particle concentration measured in the sea spray simulation chamber was higher during the Oceania campaign than
during the Electra campaign (probability value of $p = 0.00$ and at a significance level of 5 %). However, when comparing the



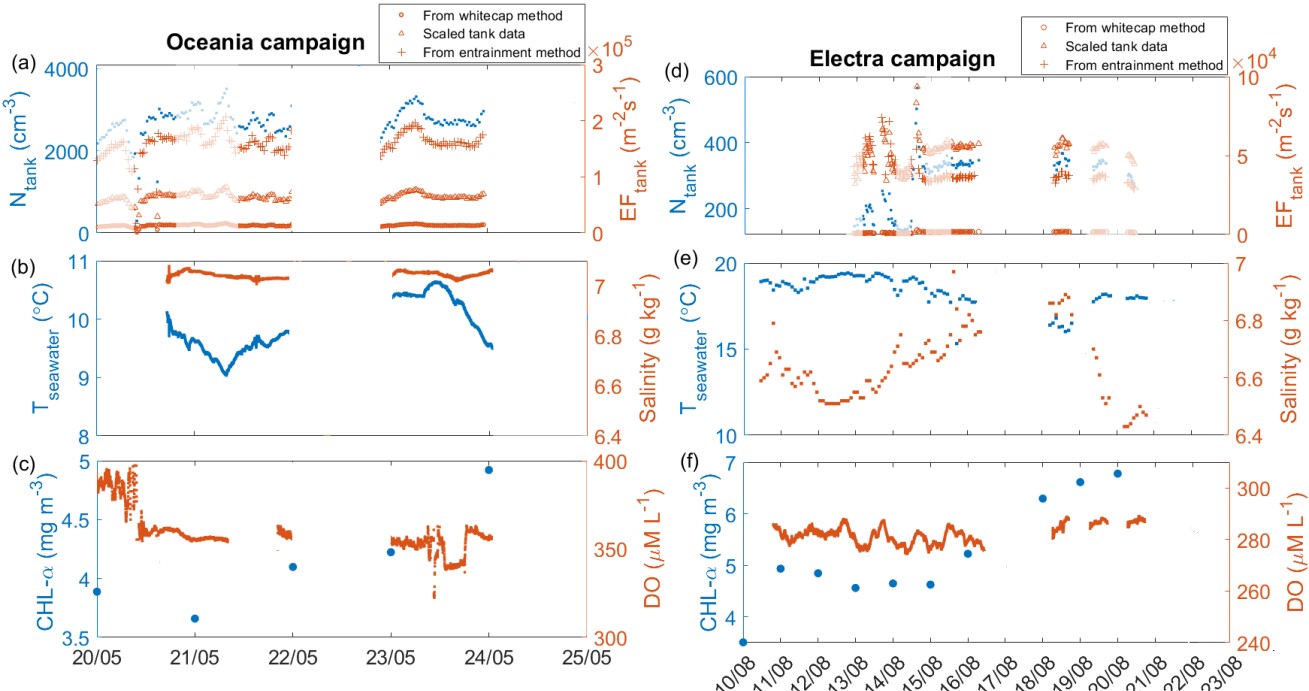

**Figure 5.** This figure shows time series of various measurements from the sea spray simulation chamber during the Oceania and Electra campaigns. Panel (a) displays the particle concentration measured in the headspace of the sea spray simulation chamber, along with flux estimates derived from the entrainment method, the continuous whitecap method, and from scaling the chamber data to the in situ fluxes. Shaded periods indicate when the ship was not anchored close to Östergarnholm or when the wind was blowing from outside the open sea sector. Panels (b) and (c) show the seawater temperature and salinity and the concentrations of chlorophyll-$\alpha$ and dissolved oxygen for the Oceania campaign, respectively. Panels (d)-(f) display the same measurements as panels (a)-(c), but for the Electra campaign.

data from the two campaigns, it is important to note that the experiments during the Oceania campaign were run at a higher jet flow rate (3.5 L min$^{-1}$ compared to 1.3 and 2.6 L min$^{-1}$ during the Electra campaign). The sudden increase in particle concentration on 14 August was due to an increase in the plunging jet flow rate from 1.3 to 2.6 L min$^{-1}$. Another factor that may have contributed to the higher particle concentration measured during the Oceania campaign is the lower seawater

temperatures in May (around 10°C) compared to August (around 17°C). Previous studies have observed an increase in particle production at lower seawater temperatures (e.g. Woolf et al., 1987; Bowyer et al., 1990; Mårtensson et al., 2003; Sellegri et al., 2006; Zábori et al., 2012; Salter et al., 2014, 2015; Nielsen and Bilde, 2020; Zinke et al., 2022). In contrast, other studies (e.g. Schwier et al., 2017; Forestieri et al., 2018) have reported an increase in particle production with increasing seawater temperatures.

Salinity was fairly constant during both campaigns (6.4-7 g kg$^{-1}$). As the solubility of oxygen in water decreases with increasing temperatures, it is not surprising that the dissolved oxygen concentrations during the May campaign were higher than during the August campaign. Additionally, the concentration of chlorophyll-$\alpha$ was higher during the August campaign.



Chlorophyll-$\alpha$ is often used as a proxy for biological productivity, which in turn can influence the concentration of dissolved oxygen through photosynthesis and respiration.

Figure S13 in the supplement shows the mean number size distribution and total concentration of SSA particles in the headspace of the sea spray simulation chamber measured by DMPS and WELAS at different jet flow rates during both campaigns. It is evident from this comparison that the total number of particles produced in the sea spray chamber increased with increasing jet flow rate, while the size distribution remained constant at each respective jet flow rate, with a mode centred at $\sim$100 nm and a second mode with smaller magnitude centred at $\sim$500 nm. This aerosol size distribution is similar to the size 475 distribution of inorganic sea salt measured with the same experimental set-up at $S = 6\,\mathrm{g\,kg^{-1}}$ (Zinke et al., 2022).

For the range of particle sizes where both DMPS and WELAS conducted measurements with 100 % counting efficiency (i.e. between 0.3 and 0.8 μm dry diameter), the measurements were found to be in good agreement. This justifies our decision to combine the data from the two instruments at a dry diameter of 0.35 μm.

In a previous study, Hultin et al. (2010) used a sea spray simulation chamber that was similar to the one used in this study, but 480 smaller. They also continuously replaced the seawater in their chamber with fresh local seawater and observed a dependence of the SSA size distribution measured in the headspace of their chamber on wind speed and dissolved oxygen concentration. Following their example, we investigated whether wind speed and dissolved oxygen saturation could potentially influence the size distribution and overall concentration of SSA produced in our chamber. To do so, we binned the data into three wind categories (0-5 m s$^{-1}$, 5-10 m s$^{-1}$ and $> 10$ m s$^{-1}$) and with respect to dissolved oxygen into subsaturated (DO$< 98\,\%$), 485 saturated ($98 < $ DO $< 102\,\%$) and supersaturated (DO $> 102\,\%$) seawater.

Contrary to Hultin et al. (2010), we observed no significant differences in the size-resolved particle concentration at different wind speeds ($p > 0.7$ at a significance level of 5 %) or varying DO saturations ($p > 0.96$ at a significance level of 5 %) (see also Fig. S14 and S15). We only observed a weak positive correlation between wind speed and total particle concentration for the Electra campaign ($r = 0.22$, $p = 0.14$ at $Q_{\mathrm{jet}} = 1.3\,\mathrm{L\,min^{-1}}$ and $r = 0.28$, $p = 0.009$ at $Q_{\mathrm{jet}} = 2.6\,\mathrm{L\,min^{-1}}$) but no 490 significant correlation for the Oceania campaign (r=0.01, p=0.89). Moreover, we observed only a weak negative correlation between the total particle concentration in the headspace of the simulation chamber and the concentration of chlorophyll in the seawater ($r = -0.23$, $p = 0.16$ at $Q_{\mathrm{jet}} = 1.3\,\mathrm{L\,min^{-1}}$ and $r = -0.16$, $p = 0.14$ at $Q_{\mathrm{jet}} = 2.6\,\mathrm{L\,min^{-1}}$) and a weak positive correlation between the total particle concentration in the headspace of the simulation chamber and the concentration of FDOM in the seawater ($r = 0.23$, $p = 0.15$ at $Q_{\mathrm{jet}} = 1.3\,\mathrm{L\,min^{-1}}$ and $r = 0.16$, $p = 0.14$ at $Q_{\mathrm{jet}} = 2.6\,\mathrm{L\,min^{-1}}$) during the Electra 495 campaign. Unfortunately, we did not have sufficient data points of chlorophyll-$\alpha$ and FDOM concentration for the Oceania campaign to derive a correlation. Scatterplots for these parameters versus particle concentration are shown in Fig. S16 in the supplement.

## 3.4 Scaling the sea spray simulation chamber measurements to aerosol emission fluxes

We used three different approaches to convert the particle concentration measured in the headspace of the simulation chamber 500 to emission fluxes. The first approach involved using the CWM (described in detail in section 2.7.2), while the second approach used air entrainment measurements to derive SSA emission fluxes (explained in section 2.7.3). The third approach, which we



adapted from Nilsson et al. (2021), involved scaling the particle concentrations measured in the simulation chamber headspace to the in situ emission fluxes in the particle size range where both the WELAS and Grimm OPC used in the EC flux system conducted measurements (detailed in section 2.7.4).

To calculate the average scaling factor $R_{\text{EF}_{\text{insitu}}:\text{SSSC}}$ for all size bins $0.25 < D_p < 2.5$ µm, it is necessary to have a similar slope between the chamber headspace number size distribution over $\overline{N_j'}$ and the flux distribution over $\overline{N_i'w'}$. To test for this similarity, we conducted a Kolmogorov-Smirnoff test (Massey Jr, 1951) on the particle size range between $0.32 < D_p < 0.75$,µm. The test revealed that the slopes were not significantly different at a probability value of $p = 0.93$ and at a significance level of $5\%$.

Scaling the sea spray simulation chamber data to the in situ fluxes we measured in this study allowed us to scale the concentration $c_X$ of any scalar $X$ measured in the sea spray simulation chamber air to the emission fluxes $EF$ using the following equation:

$$EF_{\text{scaled}} = c_X \times R_{\text{EF}_{\text{in situ}}:\text{SSSC}} \tag{13}$$

Examples of this could include the mass emission of compounds collected on filters connected to the sea spray simulation chamber or the number of sampled bacteria. Using this scaling factor, they could be scaled to mass emission ($\text{g m}^{-2}\,\text{s}^{-1}$) or number emission fluxes (bacteria cells $\text{m}^{-2}\,\text{s}^{-1}$).

It is important to note that the scaling factor $R_{\text{EF}_{\text{in situ}}:\text{SSSC}}$ is specific to each sea spray simulation chamber and cannot be applied to another chamber, as the experimental setup will vary depending on factors such as the flow rate of the plunging jet and the chamber dimensions.

Since we used different plunging jet rates during the Oceania campaign ($3.5\ \text{L min}^{-1}$) and Electra campaign (1.3 and 2.6 $\text{L min}^{-1}$), we had to derive separate scaling factors for each jet flow rate. Figure S17 in the supplement shows how the scaling factor depends on the jet flow rate. Furthermore, since in situ fluxes were measured at ambient RH ($\sim 80\%$ on average), while particles produced in the chamber were dried before being measured, we have converted all diameters to radii at RH=$80\%$.

Despite the good agreement of the slopes between $0.32 < D_p < 0.75$ µm, we would like to draw the reader's attention to the disparity between the emission fluxes derived from in situ measurements and the scaled chamber data at $R_{80} > 0.4$ µm. At $R_{80} > 0.4$ µm, the scaled chamber data yields emission fluxes that are higher than the emission fluxes derived from in situ measurements.We cannot entirely exclude the possibility of wall effects in the chamber experiments, particularly at high jet flow rates. In an ideal sea spray simulation chamber, all bubbles would burst without interacting with the chamber walls. However, in the current study, although the dimensions of the chamber are such that most bubbles burst without interacting with the walls, some bubbles are likely to have been influenced by the walls. One possible effect of these wall interactions is that the lifetime of bubbles interacting with the walls is reduced. Simply put, they burst upon impact with the walls instead of remaining on the water surface, potentially reducing the coalescence of bubbles at the water surface. It is possible that reduced coalescence would cause bubbles to burst when they are slightly smaller but more numerous than if there were no walls and the



bubbles were allowed to coalesce and form larger but fewer bubbles. It is more difficult to ascertain, however, how this effect
could impact the size and number of aerosols produced.

Figure 6 compares the fluxes derived from chamber experiments using the continuous whitecap method, air entrainment
measurements, and simple scaling with the EC fluxes measured on Östergarnsholm island. The fluxes obtained from the scaled
chamber data agree well with the flux estimates from the entrainment method at the two lowest jet flow rates, whereas the
continuous whitecap method underestimates the fluxes in comparison to those derived from the in situ measurements, the air
entrainment approach, and the simple scaling approach.

Since we assumed that $100\%$ of the water surface in the simulation chamber was covered with bubbles, which certainly
overestimates the actual bubble coverage in the chamber, we estimated the fraction of the water surface in the chamber needed
to yield fluxes comparable to the measured in situ fluxes. For the Oceania campaign, where the plunging jet was run at a flow
rate of $3.5\,\mathrm{L\,min^{-1}}$, we estimate that $21\%$ of the water surface would need to be covered. For the Electra campaign, where the
plunging jet was run at flow rates of $1.3\,\mathrm{L\,min^{-1}}$ and $2.6\,\mathrm{L\,min^{-1}}$, we estimated that bubbles would need to cover $2\%$ and $3\%$
of the water surface, respectively.

In previous experiments with artificial seawater at a salinity of $S = 35\,\mathrm{g\,kg^{-1}}$, $T_{\mathrm{seawater}} = 20°\mathrm{C}$ and $Q_{\mathrm{jet}} = 1.75\,\mathrm{L\,min^{-1}}$
(Salter et al., 2014), we used a wide-angle lens to take pictures of the surface of the water inside the chamber. From these
pictures, we estimate that $\sim 6\%$ of the surface of the water was covered with bubbles. Since these pictures were taken at higher
salinities, where we expect more and smaller bubbles, it is likely that this estimate is slightly higher than what we would expect
at $S \approx 7\,\mathrm{g\,kg^{-1}}$. Thus, the above-derived estimates for the percentage of the water surface covered in bubbles at the two lowest
jet flow rates used during the Baltic Sea campaigns seem reasonable.

### 3.5 Comparison of scaled chamber data and in situ emission fluxes to previous studies

Figure 6 also illustrates the comparison between the scaled chamber fluxes, in situ emission fluxes, and the existing sea spray
parameterizations by Mårtensson et al. (2003) and Salter et al. (2015) ($U_{10\,\mathrm{m}} = 6\,\mathrm{m\,s^{-1}}$ and $T_{\mathrm{seawater}} = 15°\mathrm{C}$). Both param-
eterizations show reasonably good agreement with the in situ data, with slightly higher values from the Mårtensson param-
eterization and slightly lower values from the Salter parameterization. These parameterizations were derived from chamber
experiments with artificial seawater at a salinity of 33 and $35\,\mathrm{g\,kg^{-1}}$, respectively. However, Zinke et al. (2022) reported an in-
crease in aerosol particle production at lower salinities ($6\text{-}8\,\mathrm{g\,kg^{-1}}$) relevant to the Baltic Sea, where these measurements were
conducted, compared to higher salinities ($\sim 35\,\mathrm{g\,kg^{-1}}$). The authors attributed this to an increased number of large bubbles at
lower salinities, which tend to produce numerous small film drops. The only previous sea spray aerosol flux measurements
from the Baltic Sea were conducted by Nilsson et al. (2021), which show emission fluxes that agree well with the emission
fluxes derived from this study.

In their study, Nilsson et al. (2021) attempted to scale co-located chamber experiments to EC flux measurements using the
same approach employed in the current study. However, they were unable to derive a scaling factor between the chamber
measurements and in situ fluxes due to differences in the slopes of the size distributions resulting from both methods. This
discrepancy may have been due to the inclusion of fluxes obtained from sectors with short fetch and shallow waters. The



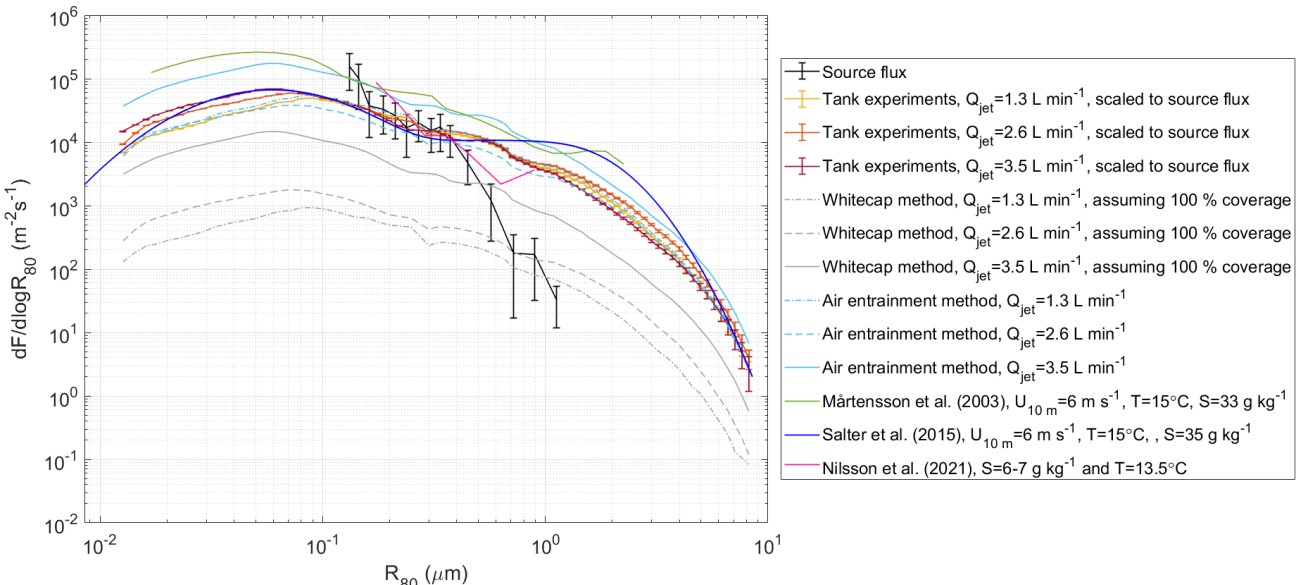

**Figure 6.** This figure compares the in situ aerosol emission fluxes measured on Östergarnsholm island with simulation chamber measurements that are scaled to the in situ fluxes, and the fluxes estimated using the continuous whitecap method and air entrainment measurements. The scaled fluxes from the chamber measurements are presented as mean with standard error, while the EC-derived aerosol emission fluxes are presented as mean with random error and discrete counting error. For comparison, EC-derived aerosol fluxes from Nilsson et al. (2021) and SSA parameterizations from Mårtensson et al. (2003) and Salter et al. (2015) are also included.

success of the current data set in this regard is likely attributed to the use of a large, homogeneous data set that is clearly defined as open-sea with a long fetch.

## 3.6 Wind speed and wave state dependent parameterizations of the scaled chamber data

In sections 3.2.2 and 3.2.3, we discussed the dependence of SSA emission fluxes on both wind speed and wave state. In this section, we have developed parameterizations of the aerosol emission flux as a function of wind speed and wave Reynolds number, which takes into account wave height, friction velocity and seawater viscosity, which in turn depends on seawater temperature and salinity. Both parameterizations are valid for seawater temperatures between 10–20°C and salinities between 6–7 g kg$^{-1}$, which represent large parts of the Baltic proper during the summer half of the year. We used scaled chamber data that encompasses dry particle diameters $0.015 < D_p < 10$ µm as a basis for the parameterizations. To parameterize the emission flux, we fit the scaled chamber data (binned based on wind speed or wave Reynolds number) to the sum of three log-normal distributions of the form:



$$\frac{d(EF)}{d\log D} = \sum_{i=1}^{k} \frac{EF_i(ReH_w \text{ or } U_{10\,\mathrm{m}})}{\sqrt{2\pi}\ln\sigma_i}\exp\left(-\frac{1}{2}\left(\frac{\ln\left(\frac{D_p}{D_{mod,i}}\right)}{\ln\sigma_i}\right)^2\right) \tag{14}$$

580    In the wind speed-dependent parameterization, the magnitude of each mode is parameterised by a log-linear relationship. For the wave state-dependent parameterization, we adopted a similar approach to that used by Ovadnevaite et al. (2014). Table 4 provides the modal diameters ($D_{\mathrm{mod,i}}$), geometric standard deviations ($\sigma_i$), and log-linear relationships for the magnitude $EF_i$ for each mode. Figures S18 and S19 in the supplement illustrate how the derived relationships fit the modes of the scaled chamber data with increasing wind speeds and wave Reynolds number, respectively.

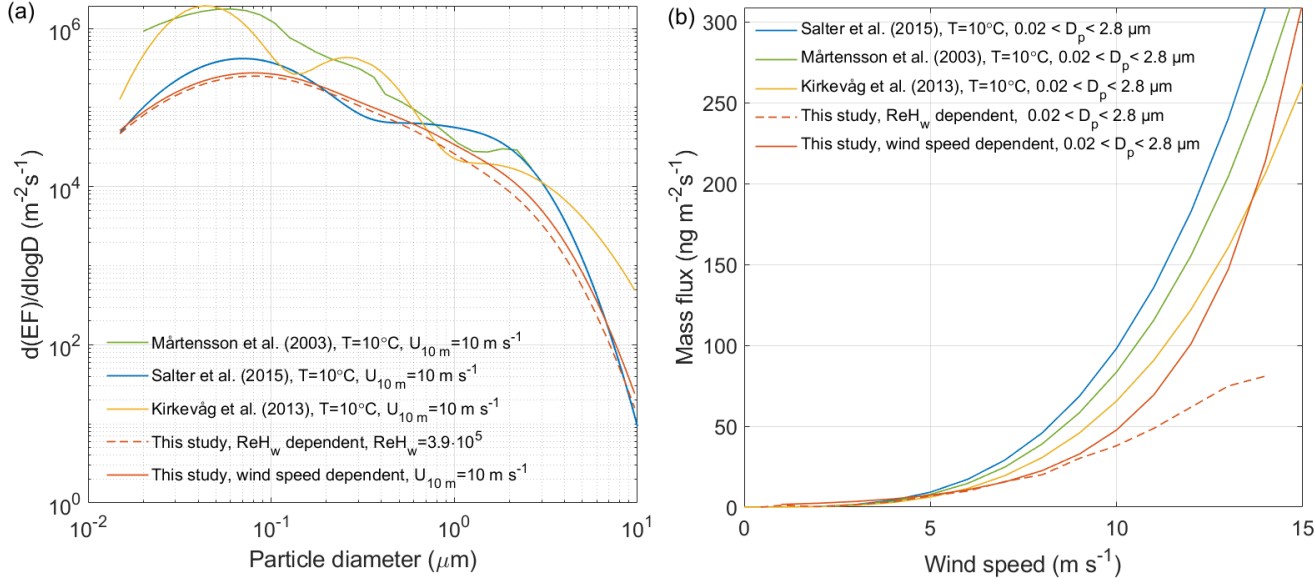

**Figure 7.** Comparison of the wind speed and wave state dependent parameterizations derived from this study with those from studies, including Mårtensson et al. (2003), Kirkevåg et al. (2013) and Salter et al. (2015). Panel (a) shows emission estimates at $U_{10\,\mathrm{m}}$=10 m s$^{-1}$, while panel (b) shows estimated mass emission flux for particles with dry diameters $0.02 < D_p < 2.8$ μm, which is the range in which the Mårtensson et al. (2003) parameterization is valid.

585    Figure 7 shows that the wind speed-dependent parameterization derived in this study produces size-resolved number emission fluxes and mass emission estimates that agree well with those obtained from the parameterizations by Mårtensson et al. (2003), Kirkevåg et al. (2013) and Salter et al. (2015). Although recent studies suggest that sea state is a better predictor of SSA emissions than wind speed alone (Norris et al., 2013; Ovadnevaite et al., 2014; Yang et al., 2019), our wave Reynolds number-dependent parameterization yields lower mass emission fluxes than the wind speed dependent parameterizations, particularly at wind speeds above $10\,\mathrm{m\,s^{-1}}$. When compared to the mass estimates from the in-situ EC flux measurements over the measured size range $0.25 < D_p < 2.5$ (see Fig. S20), it appears that these are significantly lower than the values obtained



**Table 4.** The modal diameters $D_{\mathrm{mod,i}}$, geometric standard deviations ($\sigma_i$), and log-linear relationships for the number fluxes ($EF_i$) of each of the three log-normal modes in the parameterization derived in this study.

|  | $D_{\mathrm{mod,i}}$ | $\sigma_i$ | $EF_i(U_{10\,\mathrm{m}})$ | $EF_i(ReH_w)$ |
|---|---|---|---|---|
| Mode 1 | 0.095 | 2.6 | $\log(EF_1) = 0.15 \cdot U_{10\,\mathrm{m}} + 4.3$ | $EF_1 = 0.65 \cdot (ReH_w - 1.23 \cdot 10^4)^{1.065}$ |
| Mode 2 | 0.5 | 1.72 | $\log(EF_2) = 0.18 \cdot U_{10\,\mathrm{m}} + 2.86$ | $EF_2 = 0.07 \cdot (ReH_w - 2.072 \cdot 10^4)^{1.026}$ |
| Mode 3 | 1.3 | 1.75 | $\log(EF_3) = 0.16 \cdot U_{10\,\mathrm{m}} + 2.71$ | $EF_3 = 0.02 \cdot (ReH_w - 1.162 \cdot 10^4)^{1.043}$ |

from the wind speed dependent parameterization and that they agree better with the values obtained from the wave Reynolds number-dependent parameterization. The parameterizations by Mårtensson et al. (2003), Kirkevåg et al. (2013) and Salter et al. (2015) were developed for high salinity conditions. Thus, it is reasonable to expect lower mass production in the sea spray simulation chamber at lower salinities ($S \approx 7\,\mathrm{g\,kg^{-1}}$), such as those encountered in the Baltic Sea (Zinke et al., 2022).

## 4 Summary and conclusion

In this study, we compared SSA production fluxes derived from sea spray simulation chamber measurements and in situ EC fluxes measured close to the ICOS station on Östergarnsholm island during two ship-based campaigns in May and August 2021. By combining these data sets, we quantified the magnitude and size-resolved spectrum of SSA fluxes using fast EC flux measurements across the full range of particle sizes relevant for SSA emissions. During the two campaigns, we observed a log-linear relationship between the total in situ emission fluxes and wind speed, a power-law relationship between the total emission fluxes and significant wave height, and a linear relationship between the total emission fluxes and wave Reynolds number, similar to what has been reported in several previous studies. In contrast, we did not observe any significant impact of wind speed or dissolved oxygen concentration on the size-resolved particle production in the sea spray simulation chamber experiments, as reported in previous studies. We only observed a weak negative correlation between the particle production and the concentration of chlorophyll-$\alpha$ and a weak positive correlation between the particle production and the concentration of FDOM in the seawater.

We were able to scale the chamber measurements at three different jet flow rates to obtain realistic emission fluxes using three different approaches: 1) the continuous whitecap method, 2) measurements of air entrainment, and 3) scaling the chamber measurements to the in situ emission fluxes. The measured size-resolved emission fluxes from this study agreed best with the emission estimates from the entrainment method at the two lowest jet flow rates, while the continuous whitecap method underestimated the emission flux by 1-2 orders of magnitude (depending on jet flow rate). This underestimatation is probably due to the assumption that $100\,\%$ of the water surface in the chamber was covered with bubbles, which was likely much lower during our experiments ($\sim 2-21\%$ depending on jet flow rate). The measured in situ fluxes and scaled chamber data also agreed well with previous flux measurements from the Baltic Sea (Nilsson et al., 2021) and the parameterizations by Mårtensson et al. (2003) and Salter et al. (2015).



Finally, we derived wind-dependent and wave state-dependent parameterizations of SSA emissions at low salinities representative of the Baltic Proper. The number and mass emission estimates derived from the wind speed-dependent parameterization are in good agreement with previous studies, while the wave state-dependent parameterization yields lower mass emission estimates. We attributed this difference to the lower salinity of the Baltic Sea and the fact that the Baltic Sea is a semi-enclosed sea and might not always be representative of open-ocean conditions.

The combination of laboratory experiments and EC measurements in this study is crucial for bridging the gap between in situ and laboratory estimates of SSA emission fluxes. This has significant implications for several reasons. Firstly, laboratory estimates of SSA emission fluxes cover the entire range of aerosol particle sizes produced by bursting bubbles. However, the accuracy of laboratory systems in replicating the wave-breaking process is still uncertain. Nevertheless, the reasonably good agreement between laboratory emission estimates using the air entrainment scaling and the in situ fluxes suggests that this approach can provide realistic estimates of SSA production. Secondly, certain aerosol types and properties cannot be effectively measured at the high frequencies required for EC measurements. For instance, the EC approach is inadequate for accurately estimating bacteria fluxes from the ocean to the atmosphere. On the other hand, laboratory systems are capable of measuring bacteria fluxes. Therefore, combining laboratory measurements with EC measurements allows us to derive realistic estimates of bacteria flux.

Based on these findings, our future work will involve utilizing multi-year EC measurements to investigate seasonal cycles in SSA emission fluxes from a coastal site in the Baltic Sea. Our focus will be particularly on emissions of bioaerosols contained within SSA. By integrating measurements from both the laboratory sea spray chamber and EC techniques, our aim is to develop a comprehensive understanding of SSA fluxes and the environmental factors influencing them.

*Data availability.* The data from this study is available at the Bolin Centre for Climate Research Database [DOI will be inserted after the review].

*Author contributions.* EDN, JZ, MS and PZ designed the experiments. EDN, PM and MM maintained the EC flux station. JZ and PZ carried out the chamber experiments. The data analysis was conducted by JZ with the help of EDN. JZ, EDN and MS prepared the manuscript with contributions from all co-authors.

*Competing interests.* At least one of the co-authors is a member of the editorial board of Atmospheric Chemistry and Physics.

*Acknowledgements.* The CROISSANT project is financed by the Swedish Research Council project 2018-04255. The aerosol flux tower and system have been financed initially by FORMAS, project 2007-1362, and have since received contributions from several FORMAS and VR projects. This research was also supported by the Polish National Agency for Academic Exchange under the Bekker Program (Decision





PPN/BEK/2019/1/00043/DEC/1) and VR under project 2016-05100. We also acknowledge financial support from the Bolin Climate Centre at Stockholm University, and logistic support from Uppsala University. The ICOS station at Östergarnsholm is maintained by Uppsala University and VR. The Institute of Oceanology, at the Polish Academy of Science, who enabled us to use their research vessel, s/y Oceania, anchoring the ship near Östergarnsholm for several days to conduct our sea spray simulation experiments onboard the ship, close to the aerosol flux tower at Östergarnsholm. Furthermore, we thank the crew and captain of R/V Oceania and R/V Electra and the technical staff

at ACES, Stockholm University, for their support. We are grateful to Marcin Stokowski for sharing the CTD and dissolved oxygen data for the Oceania cruise with us. The wave data were kindly provided by Dr. Heidi Pettersson from the Finnish Meteorological Institute (FMI). We also acknowledge the NOAA Air Resources Laboratory (ARL) for providing the HYSPLIT transport and dispersion model and READY website (https://www.ready.noaa.gov) used in this publication.

## Appendix A: Loss estimates and systematic error corrections

### A1  Fluctuation attenuation due to air transport in the tubes of closed path systems

Attenuation of turbulent fluctuations in the sampling tube causes an underestimation of the EC flux ($\delta_{asl}$). Co-spectral power frequency transfer functions can be used to estimate the corresponding loss of particle fluctuations in laminar flow (Lenschow and Raupach, 1991). Following Ahlm et al. (2010) and Horst (1997), we used a first-order response time of 0.6 s to estimate the damping in the sampling lines and the OPC, which will be described in detail in the next section. The aerosol EC fluxes

were corrected as follows:

$$(\overline{N'_X w'})_c = (\overline{N'_X w'}) + \delta_{asl} \tag{A1}$$

### A2  Losses in aerosol fluxes due to the limited response time of the OPC

To accurately measure EC fluxes, instruments must have a time resolution of 10-20 Hz to capture the smallest eddies. However, some instruments like OPCs are unable to achieve this resolution, resulting in significant flux attenuation. To correct for this,

we used the equations for the atmospheric surface layer from Horst (1997). By solving the integral of transfer functions and co-spectra analytically, they derived the flux attenuation ($F_a$) as follows:

$$F_a = \frac{(\overline{N'_X w'})}{(\overline{N'_X w'})_c} = \frac{1}{[1 + \frac{2\pi n_m \tau_c \overline{U}}{z_m}]^\alpha} \tag{A2}$$

Here $(\overline{N'_X w'})_c$ is the ideal flux that would have been measured if the sensors response time was not too long (where $N_X$ is either $N_i$ or $N_{\text{total}}$), $n_m$ is the dimensionless frequency at the co-spectral maximum and is a function of atmospheric stability,

$\tau_c$ is the instrument's first order response time, $\overline{U}$ is the mean wind speed at measurement height $z_m$. Here $\alpha = 1$ for stable stratification and $\alpha = 7/8$ for neutral and unstable stratification. The normalized frequency $n_m$ can be estimated for stable stratification ($\frac{z_m}{L} > 0$) as follows:



$$n_m = 2 - \frac{1.915}{1 + 0.5(\frac{z_m}{L})} \tag{A3}$$

For neutral and unstable conditions, $n_m = 0.085$. The first-order response time $\tau_c$ should be determined experimentally
following Buzorius et al. (2001) and Buzorius et al. (2003). Ahlm et al. (2010) determined $\tau_c$ to be 0.3 s for the Grimm 1.109
OPC. This allows us to calculate the systematic error of limited response time $\delta_{lrt}$ as:

$$\delta_{lrt} = \overline{N_X' w'} \left( \frac{2\pi n_m \tau_c \overline{U}^\alpha}{z_m} \right) \tag{A4}$$

Therefore, the corrected flux is:

$$(\overline{N_X' w'})_c = (\overline{N_X' w'}) + \delta_{lrt} \tag{A5}$$

## A3 Aerosol particle losses within the sampling line

Particle losses due to Brownian diffusion, impaction, interception, and sedimentation in the sampling line were estimated using
the particle loss calculator (von der Weiden et al., 2009). Since the OPC mostly measures particles in the accumulation mode,
Brownian diffusion was small within the OPC range. The sampling lines were arranged vertically to minimize deposition
losses. Therefore, the corrected aerosol flux for each size bin is:

$$(\overline{N_i' w'})_c = (\overline{N_i' w'}) + \delta_{tpl}(i) \tag{A6}$$

where $\delta_{tpl}(i)$ is the corrected size-dependent tube particle losses, and the corrected total aerosol flux for the entire OPC size
range is:

$$(\overline{N_\text{total}' w'})_c = \sum_{(i=1)}^{15} ((\overline{N_i' w'}) + \delta_{tpl}(i)) \tag{A7}$$

## A4 Webb correction

The Webb correction (Webb et al., 1980), also known as the WPL correction after all three co-authors, is required because
fluctuations in temperature and humidity can cause fluctuations in scalar concentrations that are not related to the trace gas flux
to be measured. For a scalar X, such as gas or particles, with an average concentration $\overline{c_X}$, the Webb correction can be written
as

$$\delta_X^\text{Webb} = \mu \overline{c_X} \frac{\overline{c_{H_2O}' w'}}{(\overline{\rho_d}} + (1 + \mu \frac{\overline{c_{H_2O}}}{\rho_d} \overline{c_X} \frac{\overline{T' w'}}{\overline{T}} \tag{A8}$$



Here, $\mu = \frac{m_d}{m_v}$ is the ratio of molar masses of dry air and water, $\overline{\rho_d}$ is the dry air density, $\overline{c_{H_2O}}$ is the average water vapor concentration or density. We applied this correction to both $H_2O$ and $CO_2$ fluxes calculated from the Licor7500, resulting in the corrected fluxes for $CO_2$ and $H_2O$:

$$(\overline{c'_{CO_2}w'})_c = \overline{c'_{CO_2}w'} + \delta_{CO_2}^{\text{Webb}} \tag{A9}$$

$$(\overline{c'_{H_2O}w'})_c = \overline{c'_{H_2O}w'} + \delta_{H_2O}^{\text{Webb}} \tag{A10}$$

Temperature fluctuations in tubing with high thermal conductivity are reduced to $1\%$ of their initial value when the tubing length to diameter ratio $L_{\text{tube}}/D_{\text{tube}} > 600$ for laminar flow and $L_{\text{tube}}/D_{\text{tube}} > 500$ for turbulent flow (Leuning and Judd, 1996). For our OPC, which has laminar flow, the $L_{\text{tube}}/D_{\text{tube}}$ ratio is approximately 787. Thus, for the OPC, we can simplyfy the Webb correction equation to:

$$\delta_N^{\text{Webb}} = \mu \overline{N_i} \frac{\overline{c'_{H_2O}w'}}{\overline{\rho_d}} \tag{A11}$$

The corrected aerosol number flux $(\overline{N'_{\text{total}}w'})_c$ or $(\overline{N'_iw'})_c$ is now:

$$(\overline{N'_Xw'})_c = \overline{N'_Xw'} + \delta_N^{\text{Webb}} \tag{A12}$$

## A5 Kowalski correction

The Kowalski correction (Kowalski, 2001) is based on the assumption that fluctuations in the ambient saturation ratio also manifest in measurements that use a sampling line. The correction is formulated in terms of the error in the form of a perceived "false" deposition velocity ($v_d$) as a function of the kinematic fluxes of temperature and water vapor and is based on the assumption of a Junge particle size distribution (Junge, 1972). It includes a single term to describe the hygroscopicity of the particles. However, the Junge distributions are outdated today, as they assume a continuously decreasing aerosol number concentration with increasing size, which is not valid for sub-micrometer aerosols with the fine structure of accumulation mode, Aitken mode, and possibly nucleation modes. Nevertheless, the OPC size range from $0.25 < D_p < 2.5\ \mu m$ only includes the larger fraction of the accumulation mode and the coarse mode, which roughly behaves like a Junge distribution. The perceived particle deposition velocity due to fluctuations in saturation ratio and deliquescence is given by the equation:

$$\Delta v_d^{\text{Kowalski}} = -\frac{(-K_F\beta)}{3\overline{e_s}(1-\overline{S}])^2 + 3K_fe_s(1-\overline{S})}(\overline{e'w'} - \overline{e}\frac{B}{(\overline{T}+CC)^2}\overline{T'w'}) \tag{A13}$$

Here, $K_f$ is constant dependent on ability of particle ensembles to exhibit deliquescence (set to 0.5 for polluted continental air mass), , $\beta$ is the Junge power law constant (set to 3), $\overline{e_s}$ is the half-hour average water vapor saturation pressure, $\overline{S}$ is the



average saturation ratio, $\overline{e'w'}$ is the water vapor pressure vertical flux (corrected from $\overline{c'_{H_2O}w'}$), $\overline{e}$ is the average water vapor pressure, $B$ is a constant used for the saturation vapor pressure, $\overline{T}$ is the average air temperature, and $CC$ is a coefficient from the relationship between $\overline{e_s}$ and $\overline{T}$. The Kowalski error can now be written as:

$$\delta_{\text{Kowalski}} = -c\Delta v_d^{\text{Kowalski}} N_X \cdot 10^6 \tag{A14}$$

Here, $N_X$ is either the total concentration $N_{\text{total}}$ or the size-resolved concentration $N_i$ measured by the OPC. The factor $10^6$ adjusts for the conversion of $\text{cm}^{-3}$ to $\text{m}^{-3}$ in the aerosol concentration if $\Delta v_d^{\text{Kowalski}}$ is given in $\text{m s}^{-1}$, such that the flux correction is in units of $\text{m}^{-2}\text{s}^{-1}$.

## A6   Random Errors

Regardless of the method used to quantify the random flux measurement uncertainty, some characteristics of the uncertainty have been shown to be extremely robust, both with respect to different fluxes (i.e., for $H$ and $\lambda E$ as well as scalar fluxes $F_c$) and across a variety of sites and ecosystem types (Aubinet et al., 2012). The standard deviation of the random measurement uncertainty generally increases with the magnitude of the flux in question (in our case, we are interested in $[\overline{N_X'w'}]$), and this relationship can be approximated as follows:

$$\sigma(\epsilon_X^{\text{random}}) = a + b \times [\overline{N_X'w'}]] \tag{A15}$$

While $a$ may vary greatly (Richardson et al., 2008), $b$ usually varies only from 0.1 to 0.2. A consequence of the nonzero intercept, $a$, is that there is a baseline of residual uncertainty even when the flux is zero. This implies that relative errors decrease with increasing flux magnitude.

### A6.1   Discrete counting error of aerosol EC fluxes

The OPC operates by passing air flow through a laser mounted at a 90° angle to a photodiode. On the opposite side of the photodiode, there is a copper mirror that produces light reflection for the photodiode. Each size channel of the OPC has been calibrated separately in comparison to a reference instrument. When a particle passes through the laser, it creates a voltage spike in the photodiode. The height of this signal determines the particle size. Instruments like the OPC and CPCs count each particle discretely. For such instruments, a large part of the random errors is related to the discrete nature of the data. The error in the aerosol number concentration $\overline{N_i}$ is proportional to $(n)^{1/2}$, where $n$ is the total count per half-hour period, or $(n)^{1/2} = (\overline{N_i}Q\Delta t)^{1/2}$ where $Q = 1.2\,\text{L min}^{-1}$ is the sample flow of the OPC and $\Delta t = 30\,\text{min} = 1800\,\text{s}$ is the sampling period. This is often called the discrete counting error or the square root counting error. When applied to $\overline{N_i'w'}$ or $\overline{N_{\text{total}}'w'}$, the error expression $\epsilon_X^{DC}$ (for either an individual size bin i or the total OPC size range) becomes:

$$\epsilon_X^{DC} = \frac{\sigma_w \overline{N_X}}{\sqrt{\overline{N_X}Q\Delta t}} \tag{A16}$$



where $\sigma_w$ is the vertical wind variance (Fairall et al., 1983; Buzorius et al., 2003).

**Appendix B:  Diurnal cycles**

Figure B1 illustrates the diurnal variations of several variables during both campaigns, including the ambient particle concentration, in situ fluxes, wind speed, stability, friction velocity, turbulent kinetic energy, roughness length, neutral drag coefficient, wave age, significant wave height, seawater and air temperature, sensible and latent heat flux, dissolved oxygen, and chlorophyll-$\alpha$ concentration.

In general, diurnal cycles in surface layer turbulence are driven by vertical and horizontal winds, temperature, and water
vapor. The diurnal cycle of air temperature is influenced by radiation and turbulent heat fluxes. Deviations from this cycle can occur due to air advection driven by synoptic weather. The energy absorbed by the sea is distributed in the surface layer through turbulent transport in the water. This is in contrast to continental surfaces, where absorption of radiation leads to local warming and more pronounced diurnal temperature cycles. As a result, diurnal cycles in temperature are less distinct over oceans. During both campaigns, air temperature peaked during daytime, while seawater temperature remained relatively constant.

The stability remained close to neutral, with $\frac{z}{L}$ ranging from -0.1 to +0.1, as shown in Fig. S9), but it was slightly stable during the night and early morning hours (see Fig. B1b). The friction velocity, $u*$, reached a minimum value of just below 0.1 m s$^{-1}$ at midnight and a peaked at about 0.3 m s$^{-1}$ at noon (see Fig. B1c). The turbulent kinetic energy reached its maximum in the afternoon (see Fig. B1c). The diurnal cycle of the significant wave height, $H_s$, showed an increase from 0.6 m in the morning to just below 1 m in the afternoon (see Fig. B1e). The surface roughness, $z_0$, followed the same diurnal cycle as $H_s$,
with peak values of 0.16 mm in the afternoon when the highest waves were observed and the smallest roughness of 0.036 mm in the morning when the waves were smallest (see Fig. B1d). The diurnal cycle of the wave age had the opposite shape compared to $H_s$, with a median minimum from 0.7 to 0.9 from noon to afternoon and maximum median values from 1.1 to 1.5 during the rest of the day (see Fig. B1e). Therefore, the highest waves were also the youngest. Finally, the neutral drag coefficient, $CD_N$, followed the same average diurnal cycle as $H_s$ and $z_0$, with a morning minimum at $0.99 \cdot 10^{-3}$ and an afternoon peak
at $1.27 \cdot 10^{-3}$ (see Fig. B1d).

The ambient aerosol number concentrations measured on Östergarnsholm exhibit a slight increase during nighttime, although there is a large variation in the data (see Fig. B1a). This contrasts with the findings of Long et al. (2014), who observed a peak in particle concentration during daytime. Aerosols in the Baltic region can also have anthropogenic origins, originating from industries and populated areas around the coast as well as from ship exhaust. During night and morning hours, stratification
is often stable, particularly in May, which results in a shallower surface layer, less mixing and consequently higher aerosol concentrations.

The EC aerosol flux measured on Östergarnsholm exhibited a maximum during daytime, which is related to the peak in wind speed around midday (see Fig. B1a). Nilsson et al. (2021) measured EC aerosol fluxes at the Kalmar strait in the Baltic Sea and observed a similar diurnal pattern in EC aerosol fluxes and wind speed.




The concentration of dissolved oxygen during the Oceania cruise in May showed a minimum just after 12:00 and higher concentrations around 06:00 and 18:00 (see Fig. B1h). During the Electra campaign, the dissolved oxygen concentration had its minimum during the early morning hours (around 06:00) and was highest during midday. A similar diurnal cycle in dissolved oxygen concentration measured during the Electra campaign was observed by Hultin et al. (2011). The concentration of chlorophyll-$\alpha$ during the Electra campaign also showed a minimum during the early morning hours and increased slightly in the afternoon (see Fig. B1h). Unfortunately, there were not enough data points for the chlorophyll-$\alpha$ measurements during the Oceania campaign.

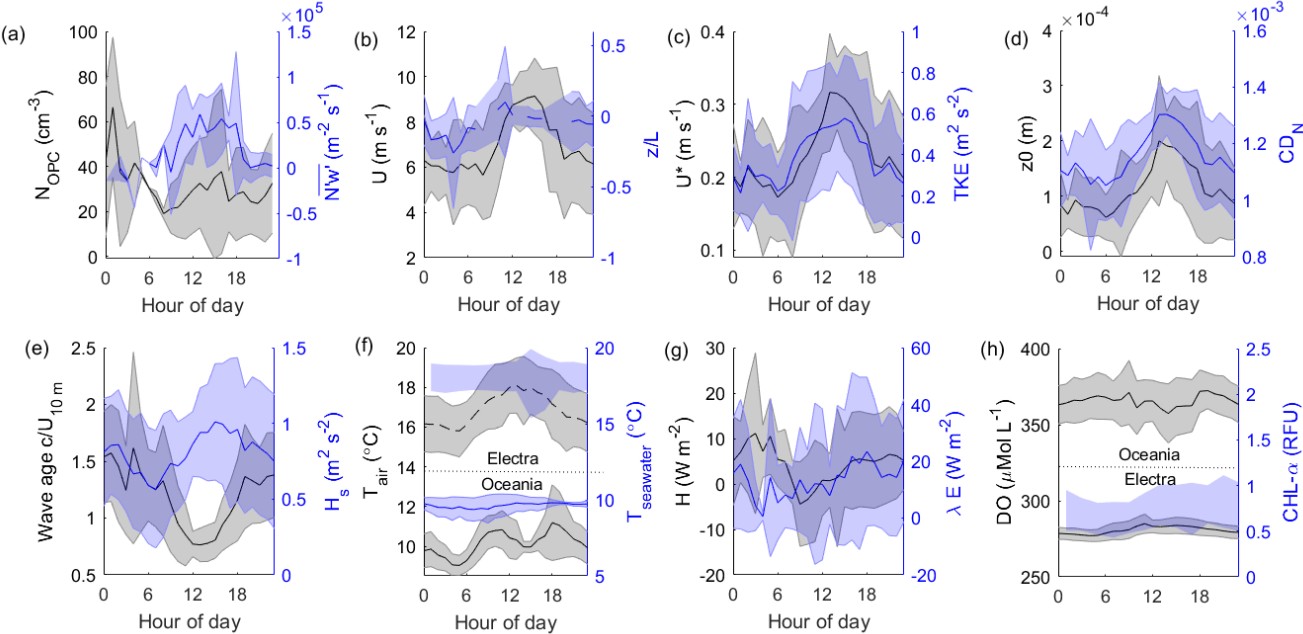

**Figure B1.** Diurnal cycles (presented as mean and standard deviation) for the Oceania and Electra campaign: (a) ambient particle concentrations and EC fluxes measured on Östergarnsholm, (b) wind speed and stability, (c) friction velocity and turbulent kinetic energy, (d) Roughness length and neutral drag coefficient, (e) wave age and wave height, (f) air temperature and seawater temperature, (g) sensible heat flux and latent heat flux and (h) dissolved oxygen and chlorophyll-$\alpha$ concentration for both campaigns. For $T_{\mathrm{air}}$, $T_{\mathrm{seawater}}$ and DO concentration, the dirunal cycles are shown separately for the two campaigns. For the Oceania campaign, there are not enough data points of chlorophyll-$\alpha$ for a diurnal cycle.



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
