# Peer review of "Sea spray emissions from the Baltic Sea: Comparison of aerosol eddy covariance fluxes and chamber-simulated sea spray emissions"

_EGUsphere, 2023_

## Author Response (AR1)

**Reply to review**

We would like to thank the reviewers for their valuable feedback and constructive comments on our manuscript. In the following sections, we present detailed responses to the comments from all reviewers. In addition to incorporating changes based on the reviewers' suggestions, we have implemented several improvements and modifications during the revision process:

1. **Aerosol EC flux correction:**

   Due to issues with the Licor 7500 on our 12 m tower, only 41% of the aerosol EC fluxes were fully corrected in the initial submission. During the review process, persistent concerns led us to supplement the dataset with values from an identical Licor 7500 (affiliated with Uppsala University) at the 10.4 m level in the 30 m Uppsala mast, located 8 m east of our tower. This adjustment enabled the correction of nearly 100% of the aerosol EC fluxes, significantly increasing the number of data points.

[Figure]

*Figure 1 Picture of the two flux towers including heights of the respective Licors.*

2. **Printing Error in the Kowalski Correction:**

   During the review process, a printing error in the Kowalski correction, specifically Equation 15 in Kowalski (2001), was identified. A missing term in this equation resulted in an incorrect unit ([K m s-1] instead of [m s-1]) for the dry deposition velocity it described. As a consequence, we have decided to omit the Kowalski correction from our analysis, given that the corrected form of the equation yielded highly unrealistic values.

3. **Sampling Line Response Time Correction:**

A discrepancy in the first-order response time for signal damping in the sampling line was found. The submitted manuscript incorrectly stated 0.6 s instead of the accurate value of 0.4 s. The correction has been made throughout the manuscript, leading to a revised flux loss of 14.8% (instead of the previously mentioned 20.7%).

4. **Figure and Table Updates:**
All figures have been updated using the refined dataset, and corrections have been applied to the net flux and emission flux estimates in Table 3. It's noteworthy that these adjustments did not impact the wind speed- and wave state parameterizations. These parameterizations continue to accurately represent the scaled fluxes from the updated dataset, exhibiting even higher coefficients of determination due to the expanded dataset.

Based the reviewer's request to shorten the manuscript, we shortened the discussion of Figure 4 by removing the comparison to fits from previous studies that were obtained from CPC measurements with the reasoning that CPC measurements are not directly comparable to OPC measurements given then different size ranges.

Reviewer 1:

Zinke et al. described the combination of aerosol eddy covariance fluxes and aerosol number size distribution from a plunging jet to estimate the emission fluxes of sea spray between 0.015 and 10 um. The method has been described by Nilsson et al., (2021). The results are interesting and useful for improving sea spray source functions, but the authors have included too many details and unnecessary comparisons, making it difficult to follow the main narrative. The paper is written by well-known experts and represents their continuing work on sea spray aerosol. I recommend that the paper be published after addressing some issues:

- The claim of bridging the gap between in-situ and laboratory estimates is a bit of an oversell. To me, bridging the gap means identifying a sea spray profile in ambient marine environments that agrees well with chamber-produced sea spray. However, the representativeness of the chamber in mimicking the true ocean remains unclear. The study simply rescales the number size distribution by DMPS to EC measurement.

We acknowledge that characterizing our manuscript as "bridging the gap" may be an exaggeration. In response, we have revised the abstract by removing the phrase "bridging the gap" and rephrasing the first sentence as follows: "To compare in situ and laboratory estimates of sea spray aerosol (SSA) production fluxes, we conducted two research campaigns [...]"

Regarding the chamber's ability to replicate open ocean conditions: Previous studies by Hultin et al. (2011), utilizing a similar but smaller chamber, and Salter et al. (2014), using the same chamber employed in our study, demonstrated that plunging jet sea spray simulation chambers generate a (surface) bubble spectrum representative of open ocean conditions.

- The paper is lengthy, and I suggest that the authors restructure it. In many cases, I had to go back and forth between the Supplementary Figures and the main text.

To maintain a concise manuscript length, we initially minimized the number of figures within the main text. Consequently, some figures were relocated to the supplementary

materials. In response to the recommendation from reviewer 2, we have transferred Figure S10 back to the main document, now designated as Figure 5.

Based on the reviewer's request to shorten the manuscript, we shortened the discussion of Figure 4 by removing the comparison to fits from previous studies that were obtained from CPC measurements with the reasoning that CPC measurements are not directly comparable to OPC measurements given then different size ranges.

- The method used in the study was first described by Nilsson 2021. In the original paper, merging DMPS and EC in the range of 0.3 to 0.5 um suffers from large uncertainty. How was this uncertainty dealt with in this study?

In the study by Nilsson et al. (2021) the uncertainty that resulted from merging the DMPS from the chamber experiments and EC measurements using the Grimm OPC was probably to a large extent caused by a limited amount of data from the open sea sector. In the current study we had enough data from the open sea sector to have a good agreement between the DMPS measurements connected to the chamber and the EC measurements using the Grimm OPC in the size range between 0.32<Dp<0.75 µm that allowed us to scale the chamber data to the EC flux measurements.

- The temporal resolution of DMPS and EC is quite different; how did the authors merge two datasets with such different resolutions?

The DMPS and WELAS data obtained from the chamber experiments was resampled to the time resolution of the EC flux measurements. However, the scaling factor was derived by comparing the averaged size distributions from the chamber experiments to the averaged EC measurements.

- Line 37: The main reason for the discrepancy is the method used to obtain sea spray source functions. For example, Liu et al., use particles > 0.5 um as a proxy for sea salt, which is not representative of sea spray source functions.

We appreciate the reviewer's comment, and in response, we have incorporated a sentence addressing the concern raised: "One reason for the discrepancy could be the method used to obtain sea spray source functions. For example, Liu et al. (2021), used particle diameters >0.5 µm as a proxy for sea salt, which is not representative of sea spray source functions"

- Line 48: Even different labs produce different sea spray source functions.

We have reformulated the following sentence to include the reviewer's suggestion: "The large variability in sea spray source functions **produced from different laboratory studies** may also be due to the different approaches used to derive them."

- Line 349: What are the ending-heights of HYSPLIT?

We have added this information to the text: "Back-trajectory models **with endpoint heights at 100 m** were computed using HYSPLIT for both campaigns"

- Line 349: The language could be more concise; everyone knows that Figure S7 is in the supplementary materials.

Based on the reviewer's comment we have removed "in the supplementary materials" from the text.

- Line 352: Is it a real seasonal trend or just two different days?

We posit that the observed trend is seasonal, supported by the consistent measurement of higher seawater and air temperatures throughout the 12-day campaign in August compared to the campaign conducted in May.

- Line 354: I do not understand why the authors spend so many words explaining the difference in chlorophyll-a concentrations that come from different seasons.

In response to the reviewer's feedback, we have revised the discussion on chlorophyll-a concentrations, streamlining the content. The updated passage now succinctly conveys: "The Oceania campaign in May fell between the spring and summer blooms, while the Electra campaign in August coincided with the late stages of the summer bloom."

- Line 358: Is it 186030 minutes or 186 hours and 30 minutes?

We believe the reviewer's comment pertains to line 378, where we originally referred to 186 half-hour periods. To enhance clarity and prevent any confusion, we have replaced "30 min" with "half-hour periods" in the text. With the updated dataset, this count has now risen to 203 half-hour periods.

- Line 407: This is probably the most prominent reason; sea spray with different size ranges is barely comparable.

We concur with the reviewer's observation. We have now removed the comparison to the CPC data from previous studies as explained in a previous reply.

- Figure 4: Why are the first and last bins so similar for different U10 or ReHw?

This can be attributed to a shortage of data points in the initial and final size bins for each U10 or ReHw with the original dataset. With the utilization of the updated dataset, we note a more significant variation in the last bins across various U10 and ReHw values.

- Line 445: The authors compared their size-resolved emission fluxes with Ovadnevaite 2014. Although the trend is similar, the shape is quite different. I don't think it 'agrees very well'.

We appreciate the reviewer's comment and, in response, have replaced "agrees very well with" with "exhibits a similar trend" for clarity.

- Line 456: Since jet flow rates are different, comparing number concentrations is meaningless.

We have omitted the initial sentence of the paragraph and revised the subsequent statement for enhanced clarity: "When contrasting the data between the two campaigns, it's crucial to highlight that the experiments conducted in the Oceania campaign involved a higher plunging jet flow rate (3.5 L min$^{-1}$ as opposed to 1.3 and 2.6 L min$^{-1}$ during the Electra campaign). Consequently, this resulted in elevated particle concentrations recorded during the Oceania campaign."

- Line 458: The impact of water temperature is expected to be minor compared to the jet flow rate.

We appreciate the reviewer's comment and have appended the following statement to the conclusion of the paragraph: "Nevertheless, considering the relatively limited range of seawater temperatures examined in this study, the influence of seawater temperature is anticipated to be minor when compared to the impact of the jet flow rate."

- Line 470: The first few bins of WELAS are unlikely to represent real sea spray and should be excluded.

We concur with the reviewer's viewpoint (as explicitly mentioned in the caption of Figure S13, now S12). In accordance with the reviewer's suggestion, we have revised the presentation to display only the WELAS data above 0.3 µm.

- Line 594: If I understand correctly, EC measurements were also impacted by sea spray production history, as evidenced by the impact of short, long fetches, and shallow waters. But for sea spray simulation chambers, seawater is purely local; this might introduce additional uncertainty when merging two datasets.

EC measurements may be influenced by wave history, with shorter fetches exhibiting different wind-wave characteristics than longer ones. Water depth also affects wind-wave traits due to interactions between surface turbulence and the seabed. Importantly, changes in fetch conditions do not imply that particles originated from farther away.

The footprint of EC measurements is primarily determined by wind speed, direction, and atmospheric stability (assuming a constant measurement height) and is not impacted by fetch variations. Consequently, particles contributing to the upward flux should have originated within the footprint of the flux tower, remaining local regardless of fetch changes. Therefore, we dispute the notion that this introduces additional uncertainty.

Reviewer 2: Mingxi Yang

Knowing the rate of sea spray production under different conditions is important for understanding the marine aerosol budget and marine clouds. This paper presents an ambitious set of observations: 1) direct sea spray flux measurements by the eddy covariance (EC) method for diameters of 0.25 to 2.5 micron from a coastal tower; 2) chamber flux measurements using seawater collected nearby from ships for diameters of 0.015 to 10 micron.

The results and discussions are mainly structured over two main aspects:

1. Assessment of previously proposed relationships between sea spray production and its drivers, namely wind speed, wave height, and wave Reynolds number. The authors found that qualitatively, those previous parameterizations are generally consistent with this dataset

2. Combining the EC flux and chamber flux to generate a sea spray production parametrization that covers the entire range of the chamber measurements (0.015 to 10 micron). Here the authors found that applying the air entrainment approach to the chamber data leads to best agreement with the EC fluxes (though for some sizes the difference is still large, e.g. half a decade), and so chose this as the basis for their new parametrization.

Overall, I think this is a very rich dataset, and I applaud the authors' efforts in trying to bring together EC and chamber measurements. I can see their argument that this may be a useful approach to determine the specific fluxes of spray components that cannot be measured fast enough to apply EC. However, I think the paper can be improved significantly. My main recommendations/comments are:

1. (abstract/intro/conclusions) What are the key uncertainties/unknowns in sea spray production prior to this work, and what are the key improvements in our understanding after undertaking this ambitious work to couple EC & chamber fluxes? I feel that currently such key messages are largely absent.

We would like to thank Mingxi for his positive reflection on our study. We address the following key uncertainties in our work: the absence, until now, of parameterizations for calculating SSA emissions from low-salinity or brackish waters, and the lack of established techniques for measuring size-segregated atmospheric fluxes using the eddy covariance method for particles smaller than 0.1-0.3 μm (depending on OPC model). To fill the gap in parameterizing SSA emissions from low-salinity waters, we developed a parameterization based on wind speed and wave conditions, utilizing our observations from the Baltic Sea.

A further challenge facing the field is that there is not an established approach for quantifying emission fluxes of specific particle classes emitted with SSA, such as organics or bacteria. In an attempt to resolve this challenge, we employed a scaling method relating chamber measurements to fluxes. This scaling factor will also be applied in related research aiming to determine bacteria fluxes from the Baltic Sea, an area where such data is currently lacking in the available literature.

To provide a succinct overview of the state of knowledge and knowledge gaps related to SSA formation factors (seawater temperature, organics/biogenic activity, salinity, and sea state), we've added the following text to the introduction:

"Seawater temperature is a significant factor impacting SSA formation; however, the specific mechanisms and nature of this influence remain unresolved. Previous studies have reported contrasting results on how seawater temperature affects SSA production. Many laboratory studies (e.g. Woolf et al., 1987; Bowyer et al., 1990; Mårtensson et al., 2003; Sellegri et al., 2006; Zábori et al., 2012; Salter et al., 2014, 2015; Nielsen and Bilde, 2020; Zinke et al., 2022) reported an increased SSA production at decreasing seawater temperatures, while some studies using real seawater (e.g. Schwier et al., 2017; Forestieri et al., 2018) reported a decrease in particle production with decreasing seawater temperature. This disparity could potentially be explained by the presence of organics and biogenic material in the real seawater, that alter the SSA production through changes in the surface tension and bubble persistence compared to inorganic salt solutions (Modini et al., 2013).

Despite numerous recent studies, the impact of biological activity on SSA production remains uncertain. Research suggests that the presence of biogenic material can influence the quantity, size, and chemical composition of newly formed SSA (e.g., Fuentes et al., 2010; Hultin et al., 2010, 2011; Prather et al., 2013; Alpert et al., 2015; Lee et al., 2015; Wang et al., 2015; Christiansen et al., 2019). However, the extent of these effects varies among studies and is likely influenced by both the type and amount of organic compounds present in seawater (e.g., Facchini et al., 2008; Quinn et al., 2014).

Salinity is another factor that adds a layer of complexity to our understanding. A number of studies have observed a shift of the modal particle diameter to larger sizes and an

increase in particle number production at higher salinities (Mårtensson et al., 2003; Russel and Singh, 2006; Tyree et al., 2007; Zábori et al., 2012, Zinke et al., 2022), while other studies (Park et al., 2014; May et al.; 2016) observed no such shift in particle size. The effect of salinity on SSA production has been linked to changes in bubble coalescence (Lewis & Schwartz, 2004; Craig et al., 1993; Slauenwhite & Johnson, 1999) as well as to effects on the length scale of the rupturing bubble film (Dubitsky et al., 2023).

Finally, the sea state has been identified as an important environmental factor influencing SSA emissions. Recent research suggests that parameters like significant wave height or wave Reynolds number provide more accurate predictions of SSA emissions compared to relying solely on wind speed (Norris et al., 2013; Ovadnevaite et al., 2014; Yang et al., 2019). This improvement is likely due to the consideration of enhanced wave breaking in shallow coastal waters within these parameters (Yang et al., 2019). However, it is important to note that the wave Reynolds number likely also incorporates the impact of seawater temperature and salinity, factors integrated through the inclusion of seawater viscosity in this parameter (Ovadnevaite et al., 2014)."

In addition, we have appended the following text to the conclusion of the introduction: "The parameterizations developed in this study mark the first of their kind for low-salinity waters. Unlike previous models that relied on a global oceanic average salinity of 35 g kg-1, our research focuses on the distinctive characteristics of low-salinity environments. Moreover, the scaling factor identified in our study provides a means to accurately quantify emission fluxes of specific particle classes associated with SSA, like bacteria, in future investigations."

In the conclusions, we believe, this is already stated in L 617ff.

2. The authors can be more constructively critical, both when comparing against previous observations/parametrizations, and also when comparing their EC fluxes with scaled up chamber fluxes. For example, that the scaled fluxes and EC fluxes have rather different size distributions (e.g. Figure 6) are mostly undiscussed.

We thank the reviewer for his comment. However, we believe that we have already addressed the variations in size distributions between the scaled chamber fluxes and EC flux measurements. In our discussion, we point out the notable agreement in slopes within the range of 0.16<R_80<0.37 µm as well as highlighting the contrasting emission fluxes obtained from in situ measurements and the scaled chamber data at R_80 > 0.4 µm: "Despite the good agreement of the slopes between 0.16<R_80<0.375 µm, we would like to draw the reader's attention to the disparity between the emission fluxes derived from in situ measurements and the scaled chamber data at R80 > 0.4 µm. At R_80 > 0.4 µm, the scaled chamber data yields emission fluxes that are higher than the emission fluxes derived from in situ measurements." Following these observations, we proceed to speculate on potential factors contributing to these disparities, such as wall effects, among others.

If the whitecap approach (assuming 100%) is found not to be appropriate for scaling up chamber fluxes, what are the implications? Should the recommendation be to always measure the bubble fraction in the chamber? Or one shouldn't use the whitecap method at all?

In response to a recommendation from reviewer 3, we have incorporated a more realistic estimate of the whitecap fraction, derived from a prior laboratory study.

The additional sentences elaborating on this adjustment can be found in section 2.7.2: "For future research, we recommend measuring both air entrainment and the fraction of the water surface within the chamber covered by bubbles to improve flux estimates through this scaling approach.

As an approximation, we estimated the fraction of the water surface covered by bubbles in previous experiments with artificial seawater at a salinity of S = 35 g kg$^{-1}$, T$_{seawater}$ = 20°C, and Q $_{jet}$= 1.75 L min$^{-1}$ (Salter et al., 2014). These authors used a wide-angle lens to photograph the water surface inside the chamber, determining that approximately 6 % of the surface was covered with bubbles. Since these photos were taken at higher salinities, with an expectation of more and smaller bubbles, we adjusted the estimate, resulting in a whitecap coverage of 2% and 3% for the Electra campaign at flow rates of 1.3 L min$^{-1}$ and 2.6 L min$^{-1}$, respectively.

Considering the Oceania campaign's significantly higher jet flow rate (3.5 L min$^{-1}$), leading to increased bubble formation, we estimate that 21% of the water surface in the chamber was covered in bubbles during this campaign.

Those whitecap coverage estimates were determined by comparing the flux that would result from 100% whitecap coverage to the magnitude of the emission fluxes derived from the EC measurements in the overlapping size range."

The measurement uncertainties (bias and random) as presented may be underestimated. In terms of bias, given the very low flow rate of the OPC, I'd be very surprised if the high frequency flux loss is as small as the authors claim to be. I urge the authors to address this by a) closely examine the N'W' cospectrum and look at high frequency flux loss, and b) verify the response time of the OPC. In terms of random error, fluctuations in aerosol number not due to sea spray production (e.g. anthropogenic influence/transport) will significantly increase the noise in the measurement. The authors can estimate random error by shifting N' and W' by a large lag (e.g. a few minutes) and then compute the covariance – the stdev in that null covariance then approximates the random error.

We have conducted an estimation of high-frequency loss in accordance with the methodology outlined by Wolf & Laca (2007). The average flux loss at high frequencies was found to be 1%. Additionally, we employed an alternative approach by comparing the N'w' cospectrum to the cospectrum of heat, resulting in a higher estimated loss of 13.9%. To provide a comprehensive overview, we incorporated the following sentence into the text: "High-frequency losses were assessed using two different methods: 1) following the approach by Wolf & Laca (2007), and 2) by comparing the N'w' cospectrum to the cospectrum of heat. The estimated high-frequency flux losses ranged from 1% to 13.9%, respectively."

Furthermore, we addressed the reviewer's comment by estimating random errors with time shifts in N' and w' of 3 and 5 minutes, leading to computed random errors of 34.7% and 41.6%, respectively. This information has been included in the manuscript: "The random error was determined to vary between 35% and 42%, obtained by shifting N' and w' by three and five minutes, respectively, and calculating the standard deviation of the computed co-variance."

Regarding the OPC's response time, we adopted a first-order response time (τ=0.3 s) as determined by Ahlm et al. (2010). This determination involved an aerosol source and a zero filter, with a curve fitted to the declining concentration to establish the response time.

Further specific comments

Abstract.  Specify the size range of EC flux measurements

In response to the reviewer's suggestion, we have incorporated information about the size range for the EC flux measurements into the abstract.

Section 2.1 the aerosol flux tower was on the island, not on the ships, right?  The first paragraph in section 2 is a bit confusing, and it's not clear what the ships contribute to the measurements at this point.  It's only later that I realized the chamber measurements were done on the ships.

In accordance with the reviewer's recommendations, we have included a specification indicating that the tower is situated on Östergarnsholm island. Additionally, to provide clarity, we introduced the following sentence in section 2 to elaborate on the chamber experiments conducted on the ships: "Throughout both campaigns, the sea spray simulation chamber was positioned aboard ships, which were stationary in close proximity to the flux footprint area of the EC flux tower."

Line 114. Specify whether ¼" is inner or outer diameter

We now specify that ¼" is the outer diameter. The inner diameter was 5.35 mm.

Line 137. A bit more info here would be useful for each, i.e. despiking (what sigma?), rotation (double rotation? Planar fit?), detrending (linear?), lags (how long?)

We appreciate the reviewer's comment and have incorporated the recommended information into the text. It now states: "The CALCEDDY LabVIEW program […] was used for eliminating spikes **exceeding six times the standard deviation**, **double rotation** of the coordinates, **linear** detrending of the data, correcting for lags **(using a lag time ranging from 0 to 9 seconds with the largest correlation between w' and N'),** and calculating covariances, averages, and standard deviations."

Line 164-167.  I'm not sure if that's true.  Coastal areas are often subjective to strong horizontal gradients in aerosol concentrations, which could contribute to the low frequency part of the signal.  Have authors looked at parameters such as dN/dt and their impact on the flux?  I suspect that for this sort of coastal regions, a lot of the variability in N isn't due to vertical transport, and could contribute to both random and systematic errors in flux.

Rutgersson et al. (2020) conducted a comprehensive examination of the different wind sectors at Östergarnsholm and concluded that the sector (80-160°) remains unaffected by coastal influences, representing conditions typical of the open sea. We also used data from the sector 160-220°, which was categorized as open-sea for physical parameters but with possibility for some more heterogeneity in e.g. biogeochemical parameters when considering the flux footprint of the site. Influence from meso-scale systems can still exist for both categories but Rutgersson et al (2020) also stated that as long as mesoscale systems have a great enough scale (in time and space) not to invalidate assumptions of stationarity and homogeneity, this does not disturb the measurements or the assumption of representativeness for open sea conditions when studying air– sea interaction processes.

As suggested by the reviewer, in the following scatterplot, we illustrate the relationship between dN/dt and the uncorrected net flux. Notably, it is observed that dN/dt does not exhibit a significant impact on the flux.

[Figure]

*Figure 2 Scatter plot of uncorrected net EC fluxes versus dN/dt.*

Line 178-185.  Why are there two estimates, 20.7% and 13.5%?  what's the difference? Is the first the 'total' attenuation?  Given the laminar flow and very low Reynolds number, I would've expected even greater flux loss.

During the peer-review process, we identified an error in our use of the instrument response time ($\tau=0.6$ s instead of the correct value of 0.4 s) when assessing losses due to signal attenuation in the sampling line. This oversight resulted in an overestimation of the losses. Further clarification on this matter is available at the beginning of this reviewer response document. Subsequently, we corrected the values, and the updated figures now show a 14.8% error attributed to signal attenuation in the sampling line (previously reported as 20.7%) and a 13.3% error resulting from the limited response time of the OPC.

Concerning the comment about the laminar flow: The Reynolds number has increased to Re=322 in the updated dataset. The main reason for the small particle losses are as follows: The OPC's size ranges from 0.25-2.5 μm diameter mostly fall in the accumulation mode (except for the largest bins). Since Brownian diffusion is insignificant in this size range and impaction and interception are small for particles with diameters smaller than 1 μm, particle losses in this size range are small.  Furthermore, the sampling lines were designed such that particle losses were minimized, sampling vertically downwards so that sedimentation losses are insignificant. In addition, the OPC's sampling inlet is pointing upward in order to minimize the bends of the sampling line. Lastly, particle losses to the tubes differ from signal losses such that laminar flow actually gives a better penetration fraction than turbulent flow.

There are two main sources of attenuation that needs to be considered: 1) OPC only outputs 1 Hz data, and thus all flux above the Nyquist (0.5 Hz) is not captured; 2) attenuation related to the response of the instrument. It's not clear from A2 that both effects have been corrected.  Both of these terms are also dependent on the dominant scales of eddies (and so dependent on wind speed).

Two main sources contribute to attenuation: one within the OPC (represented by tau) and the other within the sampling tube. We calculated the attenuation in the tube and resulting from the limited response time of the OPC using Horst's methodology (1997), outlined in sections A1 and A2. Horst's parameterization is effective above 1 Hz and considers both wind speed and stability. However, it's crucial to recognize that this approach cannot resolve the Nyquist frequency.

Have the authors looked at the cospectrum of aerosol flux (N'w') and compare it vs cospectrum of heat to see how much high frequency aerosol flux they might be missing? E.g. plot the bottom panels of S2 on a semi-log scale and see how quickly the cospectrum goes towards 0 at high frequency.

We calculated the high-frequency flux loss from the co-spectra using the method outlined in Wolf & Laca (2007), determining that the lost high-frequency flux accounts for 1% of the measured flux. Additionally, at the reviewer's suggestion, we employed an alternative approach by comparing the aerosol co-spectrum to the co-spectrum of heat, resulting in an estimated high-frequency flux loss of 13.9%. Consequently, it is reasonable to assume that the actual high-frequency flux loss likely falls within the range of 1-14%.

Also 'constant with size' is repeated 3 times.

We have eliminated one instance of "constant with size" in line 181. The remaining two phrases pertain to distinct errors, both of which are almost constant with size (except for the largest size bins). We have clarified this in the text.

Line 212. This is a very hand-wavy guess of the random uncertainty. It could be of roughly the right magnitude, but the text should reflect that the '10-20%' number is not based on actual analysis of the aerosol data.

In response to the reviewer's suggestion, we have assessed the random error by shifting N' and w' by three and five minutes. The estimated random error varies between 35% and 42%, respectively.

Section 2.6 I think the spectral analysis can be done much better. Please see comments towards the end of the document.

In response to the reviewer's suggestion, we aimed to enhance the spectral analysis as follows: We assessed high-frequency loss through two methods – 1) following Wolf & Laca (2007) and 2) by comparing the N'w' cospectrum to the cospectrum of heat. Additionally, we incorporated the comments regarding the slope in the inertial subrange, as raised by the reviewer later in the text. Moreover, in the caption of Figure S2, we clarified that different colors represent positive and negative covariances in the co-spectra.

Figure 3. does the temperature dependence in the wave Reynolds number help to explain more variability? In a recent paper about gas exchange (doi: 10.3389/fmars.2022.826421), we found that computing the wave Reynolds number at a constant number (e.g. 20 deg C) actually leads to more variability explained than computing it at the in situ temperature.

We value the insightful comment provided by the reviewer. In our study, we calculated the wave Reynolds number in a manner akin to the approach mentioned by the reviewer. This involved using consistent values for salinity and temperature—specifically, the average

salinity (6.5 g/kg) and seawater temperature (14°C) observed during our investigation. We have incorporated this additional detail in section 3.2.1, where we initially outline the methodology for computing the wave Reynolds number.

Line 658. Here the authors say the response time is 0.6 s, but on line 675, 0.3 s is stated.  The actual response time is of course dependent on the flow rate and setup. Have the authors determined the response time of their own system for these campaigns?  It's probably not easy to determine a sub-second response time when the instrument only outputs 1 Hz data.

In the initially submitted text, a typographical error incorrectly stated the first-order response time (τ) for the signal damping in the sampling line as 0.6 seconds instead of the correct value, which is 0.4 seconds. The response time of this specific OPC (that is the response time for only the instrument itself) was determined to be 0.3 seconds according to Ahlm et al. (2010). We have rectified this typo and adjusted the resulting loss estimates accordingly. Further details about these corrections are provided at the beginning of the document.

Line 705. H2O flux is also substantially dampened when sampling down a tube.  See e.g. Fig 9 in www.atmos-meas-tech.net/9/5509/2016/    Thus the H2O related Webb correction may be even smaller than what the authors have presented.

We've introduced the following sentence into the discussion of the Webb correction in section 2.4.1, where we previously highlighted the potential dampening of density fluctuations in the sampling lines: "This aligns with the observations of Yang et al. (2016), who noted a significant dampening of water vapor fluctuations in their sampling lines."

Figure S2. Within the inertial subrange, the slope should be -5/3, not -2/3. Also, inertial subrange is typically considered to be >1 Hz.  At a height of 10 m and wind speed of 5 m, this translates to a fz/U of 1*10/5 = 2.  Currently the red lines are drawn at lower frequencies than the initial subrange.

We appreciate the reviewer's observation. The slope has been corrected to -5/3, and we have also adjusted the lines to frequencies indicative of the inertial subrange.

Bottom panel, it's not clear what the two different colours indicate.

We have now added a sentence to the figure caption explaining that the different colours indicate positive and negative covariances.

Line 565. The size distributions from the EC flux and the chamber flux are clearly different, with the EC flux falling more quickly with increasing size above ~0.4 um. The authors haven't commented much on this discrepancy. Which measurement is more realistic?

We have commented on the discrepancy between EC flux and scaled chamber flux in L524ff. We have added the following sentence to section 3.4: "Since the EC method provides a direct measurement of the fluxes, those measurements should be considered more realistic"

Figure 6. is 'source flux' the mean flux derived from EC from both campaigns?  If so, indicate the mean wind speed and temperature here, and mention that the mean wind

speed from the two cruise campaigns (6.4 and 6.6 m/s) are fairly similar to the Martensson et al 2003 and Salter et al 2015 parametrizations.

In the figure caption, we've clarified that the chamber measurements were scaled to the mean emission flux from both campaigns. Additionally, in the figure legend, we replaced "source flux" with "emission flux" and incorporated the average wind speed and seawater temperature, as recommended by the reviewer. It's worth noting that the parameterizations from Salter and Mårtensson were computed using wind speeds and seawater temperatures to align with the values encountered in our current study.

What about temperature & salinity dependence in chamber? Some of these might have been presented previously; if so perhaps a sentence would be sufficient.

In a prior study (Zinke et al., 2022), we examined the influence of salinity and seawater temperature on sea spray production in the same sea spray simulation chamber used in the current experiments. The current study involved experiments with local seawater, which contained organics, potentially affecting bubble bursting behavior beyond the impacts of temperature and salinity alone. Given that the water in the chamber was continually replenished while the ships were anchored next to Östergarnsholm island, the salinity and seawater conditions in the chamber should reflect ambient conditions. Consequently, the SSA emissions in the chamber are expected to be comparable to those measured on the island.

Line 589. It's unclear how the Reynolds number based parameterization was done for different wind speeds. What was the wave height used?

The wave height employed to calculate the wave Reynolds number was obtained from a wave rider buoy, as detailed in section 2.1. To establish the Reynolds number-based parameterization across various wind speeds, we grouped the wind speed into bins and computed the average wave Reynolds number for each wind speed bin. To clarify this process, we've included the following sentence in the caption of Figure 7: " The wave state dependent parameterization is based on averaging the wave Reynolds number within the corresponding wind speed bins, using these mean wave Reynolds number values as the foundation for the parameterization."

Line 591. I don't understand how the parametrizations (wind speed or RHw dependent) are so much higher than the EC measurements even over the same size. Weren't the parametrizations developed based on these same observations?

This discrepancy likely arose from the difference between the scaled chamber fluxes and the emission fluxes determined from in situ EC measurements for particle diameters greater than 0.4 µm in the previous version of the dataset. Since we utilized the scaled chamber fluxes as a foundation for developing parameterizations, the elevated chamber fluxes in this size range would result in higher estimates from the parameterizations compared to the emission estimates derived from the measured EC fluxes. However, with the updated dataset, the mass estimate from the EC measurements appears to align more closely with the parameterizations.

Figure S12: check caption ??

We thank the reviewer for pointing this out and have replaced "??" with a reference to equation 12.

Appendix B doesn't bring much to the main arguement. Could be move to supplement. Figures S10 and S11 may be suitable for the main paper

We concur with the reviewer's suggestion and have relocated Appendix B to the supplementary materials. Additionally, Figure S10 has been moved to Section 3.2.1 in the main text. However, we opted to retain Figure S11 in the supplement due to the paper's existing length.

Reviewer 3:

The paper by Zinke et al. reports complex experiments in reconciling sea spray production and corresponding fluxes to shed new or more light into the sea spray source function development. It is a very valuable contribution and deserves publication in EGUSphere, but more work is needed to address some important comments or omissions.

The paper looks like an attempt to summarise very large experimental effort, like PhD thesis, however, I am not suggesting splitting it into several papers, because the value would be significantly diminished, and some parts of the paper do not constitute standalone paper.

The overarching comment would be the lack of judgement throughout the paper. The authors are too complacent about similarity, comparability, or consistency between different studies. If everything is consistent and repeatable, why publish in reputable journals? The authors are encouraged to make judgements and assertions which is the only way of making progress and encouraging debate.

Major comments

Line 76. Better justification of performing SSA source function experiments in the Baltic Sea should be provided given the fact that Baltic Sea has much lower salinity in comparison to the global ocean.

We appreciate the reviewer's input and have incorporated the following text at the end of the introduction: " The parameterizations developed in this study mark the first of their kind for low-salinity waters. Unlike previous models that relied on a global oceanic average salinity of 35 g kg-1, our research focuses on the distinctive characteristics of low-salinity environments."

Line 97. Given the complex bottom topography of the Baltic Sea it is important to assess wave state vs wind speed to make the measurements scalable/consistent to the global ocean. Lower salinity of the Baltic sea can be accounted for when using Reynolds number as long as the wave breaking pattern is the same – or is it? Or any other metrics demonstrating consistent wave breaking.

Rutgersson et al. (2020) extensively evaluated the influence of bottom topography on the wave field, particularly in the wind direction sector from 80 to 220 degrees. Their findings, detailed in section 4.1.4, indicate a negligible impact on the wave field and no secondary effects on atmospheric parameters. This assessment aligns with the conclusions drawn by Högström et al. (2008). Consequently, data for these wind directions are considered to accurately represent open sea conditions for physical parameters (with limited coastal effects on biogeochemical variables, see also answer to reviewer 2).

We have added the following discussion to section 3.6: "To the best of our knowledge, prior studies have not specifically examined the influence of lower salinity on wave breaking patterns. However, this factor could potentially elucidate some of the disparities observed between our EC estimates and those conducted in higher salinity waters."

Line 106. Was the 12 m height of the EC measurements sufficient to avoid coastal surf-zone? Where was flux footprint relative to the mast distance from the surf-zone? I trust all was good, but should be supported by numbers.

The flux footprint and the coastal surf-zone of the station have been examined in previous studies by Smedman et al. (1999), Högström et al. (2008), and Gutiérrez-Loza (2022). These investigations collectively confirm that the 12 m height is adequate to prevent the sampling of aerosols originating in the coastal surf zone. Section 2.3 elaborates on this topic, and for a more in-depth understanding of the flux footprint, readers are referred to the cited studies.

Line 117. 1.2lpm flow rate through the 5m 1/4" sampling line cross-section produces lag time of ~4 seconds (if I am correct). How was this lag accounted for in the EC measurements?

The lag time was determined by shifting the time series of N' and w' against each other in increments of 0-9 seconds until we identified a peak correlation, which occurred at 4 seconds.

Line 138. I wonder if arithmetic average is appropriate here, given the fact that aerosol distributions (and atmospheric parameters in general) are log-normally distributed in nature (due to fundamentals). Median and range would be the correct parameters. I keep noting this issue for many years now, but that does not get widely accepted by excusing for historical legacy. Can it be partially for this reason that 30min average EC fluxes are so noisy?

The standard practice in the eddy covariance method involves calculating the mean, as it is inherent to the covariance computation. The covariance is derived by comparing raw data to the mean to obtain fluctuations, denoted by a prime ('). Therefore, we respectfully disagree with the reviewer's suggestion to use the median and range at this stage of the calculations. Additionally, we contest the notion that SSA size distributions are log-normally distributed since accurately representing the SSA size spectrum typically requires several log-normal modes.

Regarding the reviewer's comment on data noisiness, we are unsure about the specific aspect they are addressing. It's possible they are referring to the variability in flux estimates in relation to wind speed. If this is the case, it's crucial to acknowledge that wind speed might only be one of several factors affecting aerosol emissions. For instance, wave state, seawater temperature, and the presence of organics at the seawater surface are all likely to play a role in influencing the sea spray source flux.

Line 304. Was the assumption that the entire area was covered in bubbles tested at least visually? It is very difficult to have consistent 100% coverage without foam build-up. Foaminess would also depend on the sea water organic matter. All in all, assumption of 100% coverage is very uncertain and, even more so, incorrect in the first place. More discussion is needed on this topic and then why not to assume 30-50% with the plotted uncertainty range which would be much closer to the truth. Later in conclusions the

authors note that the coverage is more like under 20%, but that is too late and too striking.

During this study, we captured images of the bubble surface. Regrettably, our equipment lacked a wide-angle lens, making it impossible to encompass the entire water surface area within the tank. Additionally, the motion of the ships posed a challenge in obtaining sharp images for precise determination of bubble coverage. Consequently, we had to resort to utilizing images from a prior experiment conducted at a salinity of 35 g/kg. In response to the reviewer's suggestion, we have now scaled the fluxes using more realistic bubble coverage estimates. The relevant information has been incorporated into Section 2.7.2 as follows: "For future research, we recommend measuring both air entrainment and the fraction of the water surface within the chamber covered by bubbles to improve flux estimates through this scaling approach.

As an approximation, we estimated the fraction of the water surface covered by bubbles in previous experiments with artificial seawater at a salinity of S = 35 g kg-1, $T_{seawater}$ = 20°C, and $Q_{jet}$= 1.75 L min-1 (Salter et al., 2014). These authors used a wide-angle lens to photograph the water surface inside the chamber, determining that approximately 6 % of the surface was covered with bubbles. Since these photos were taken at higher salinities, with an expectation of more and smaller bubbles, we adjusted the estimate, resulting in a whitecap coverage of 2% and 3% for the Electra campaign at flow rates of 1.3 L min-1 and 2.6 L min-1, respectively.

Considering the Oceania campaign's significantly higher jet flow rate (3.5 L min-1), leading to increased bubble formation, we estimate that 21% of the water surface in the chamber was covered in bubbles during this campaign.

Those whitecap coverage estimates were determined by comparing the flux that would result from 100% whitecap coverage to the magnitude of the emission fluxes derived from the EC measurements in the overlapping size range."

Figure 3. Was the Reynolds number taking into account different salinity and temperature, resulting into different viscosity of water of the Baltic Sea vs global ocean? Same question for Figure 4.

The wave Reynolds number was computed for both Figures 3 and 4 based on the average salinity (6.5 g/kg) and seawater temperature (14°C) observed during this study. Therefore, it accurately reflects the conditions present in the Baltic Sea. This additional detail has been included in Section 3.2.1 for reference.

Line 406. Was it really comparable? I beg to differ. Sometimes it feels that the authors are too focused on consistency, comparability and so on. Differences should not be afraid or dismissed, they may inform about some important overlooked aspects or mechanisms.

Based on the reviewer's request to shorten the manuscript, we have removed the comparison to of the wind speed dependence (Figure 4) to fits from previous studies that were obtained from CPC measurements with the reasoning that CPC measurements are not directly comparable to OPC measurements given then different size ranges.

Line 410. Baltic sea salinity is 5 times lower than the open ocean and, therefore, has a profound effect on the number flux. I do not see that the authors took full account of

salinity effect other than just mentioning it. If they did, please note that where appropriate.

The salinity will have impacted our measurements but its impact will be included in the measurements. As such we did not account for the salinity effect on the size distribution of the SSA produced, which is complex and not straight-forward to parameterize. For a more detailed discussion of the effect of salinity on SSA production we refer the reviewer to Zinke et al. (2022).

Line 422. In power law relationships correlation coefficient is a poor metrics as it is almost always high due to high-end values. Maybe 95% confidence interval of the power coefficient would be more informative?

In this context, we are discussing the size-resolved dependence of the emission flux on wind speed. However, it's important to note that we employ a log-linear fit to characterize this relationship. Therefore, we are uncertain about the specific power-law coefficient mentioned by the reviewer.

Line 486. Flux and particle concentration are two different concepts or measures. Flux was demonstrated to depend on the wind speed as it should be. The ambient concentration would only be dependent on the wind speed by the power law in fully developed/equilibrated boundary layer. It takes 1-2 days to fill the boundary layer with corresponding fluxes and the Baltic Sea is too small for the established air mass or synoptic scale processes. Overall, it just manifests that the wind dependent fluxes resulted in little changes in ambient concentration emphasizing that environmental impacts of small water bodies will be limited (going back to justification of the Baltic Sea experiments).

In Line 486, our discussion pertains to the wind speed dependence of particle concentration within the headspace of the sea spray simulation chamber. One would anticipate that the particle concentration in the chamber's headspace remains independent of wind speed since it is isolated from the ambient atmosphere, and the bubbles are generated by a plunging water jet with a constant flow, aligning with our observations. However, Hultin et al. (2010) noted an effect of wind speed on the produced aerosol size distribution in a sea spray simulation chamber. They attempted to explain this phenomenon by attributing it to wind-induced upwelling of organics.

Section 3.5 What do we learn from the scaling and what is the purpose of sections 3.5&3.6 other than every chamber experiment will produce different scaling and the result? Maybe I missed something, but that should be clearly explained if I did.

Scaling the chamber experiments to sea spray fluxes enables us to scale additional parameters concurrently measured in the chamber. This is especially valuable for parameters that are too slow for the application of the EC method, as mentioned in Line 515f (now L543f in the track changes document) and Line 627f (now L657f). Specifically, we intend to derive realistic bacteria fluxes using the scaling factor established in this study, with the results intended for publication in a separate paper. Additionally, one of our objectives is to expand the size-resolved flux estimate beyond the size range covered by the Grimm OPC on the flux tower.

Figure 7. Fig 7b can be easily predicted from 7a as the mass is dependent on the particle distribution >1um. Why mass is important as it varies massively moving towards super-micron range?

Calculating the mass of sea spray aerosols holds value for two primary reasons. Firstly, numerous datasets measure sea salt mass at various global locations, offering the potential for comparison with the parameterizations we have developed. Secondly, in climate model simulations, the mass of SSA produced is frequently utilized to gauge its influence on the radiation budget.

Table 4. Geometric sigma, especially of Mode 1 is way too large to be described by one mode. The fundamental rationale is that each log-normal mode is defined by individual production mechanism and typically does not exceed 1.4-1.5 (refer to Hinds, Aerosol Technology Textbook).

We have used sigmas that are close to those used in Salter et al. (2015) and Kirkevåg et al. (2013). The parameterization by Salter et al. (2015) was found to represent the chamber produced SSA well.

Line 601. Log-linear relationship is effectively power law, why confusing readers with different mathematical or plotting outcomes when there should be enough of distinguishing linear versus non-linear effects.

It's worth noting that many prior studies exploring sea spray aerosol emissions have consistently presented the dependence of emission fluxes on wind speed as log-linear relationships. Adopting this approach aligns our study with established practices, promoting better comparability across research in the field.

Minor comments

Line 376. Previous section reported 475+150=625 half-hourly fluxes.

The reviewer correctly highlights the discrepancy. In the earlier version of the dataset, the missing data points were represented as NaN values. With the updated dataset, this has transformed into 157 data points with downward fluxes and 491 data points with upward fluxes, summing up to a total of 648 data points.

Line 401. What was the power law coefficient?

We are a bit uncertain about what the reviewer is referring to, as we don't explicitly mention any power-law relationship in this line. Instead, we opted to depict the relationship between EC fluxes and wind speed as a log-linear one, aligning with the approach of many prior studies for improved comparability.

[revised manuscript text omitted]